# Beyond Examples: Constructing Explanation Space for Explaining Prototypes

## Abstract

As deep learning has been successfully deployed in diverse applications, there is ever increasing need for explaining its decision. Most of the existing methods produced explanations with a second model that explains a given black-box model, but we propose an inherently interpretable model for faithful explanation. Our method constructs an explanation space in which similarities in terms of human-interpretable features at images share similar latent representations by using a variational autoencoder. This explanation space provides additional explanation of the relationship, going beyond previous classification networks that provide explanation with distances and learned prototypes. In addition, our distance has more intrinsic meaning by VAE training techniques that regulate a latent space. With human evaluation, we validate the quality of explanation space and additional explanations.

## 1 Introduction

"*Explanation*" is essentially based on understanding. When a fact is in doubt, an explanation is provided to resolve a question. Therefore, explanation must be human-interpretable and reliable. An explanation that is not human-interpretable does not resolve the issue, and an explanation that is not reliable may create confusion. In practice, it is difficult to find an explanation that perfectly satisfies even one condition. In this situation, when an insufficient explanation is given, a use may wonder why the explanation is given. We aim to address this question by providing a reliable explanation and providing additional explanations for a given explanation in this paper.

With advances in deep learning, its performance improves rapidly. Advances have increased the use of deep neural networks (DNNs), which is used in areas where model decisions significantly affect people - such as healthcare (Caruana et al., 2015), radiation (Reardon, 2019), and autonomous driving. As a result, questions about the rationality of model predictions for industrial and social use have been raised, and the call for explanation is growing (Goodman & Flaxman, 2017; Doshi-Velez & Kim, 2017). Since most DNNs are black-box models, explanation of model prediction is often provided by a second model that explains a given black-box model (Ribeiro et al., 2016b; Chen et al., 2018a; Lundberg & Lee, 2017; Guo et al., 2018). This second model is called a post-hoc interpretable model, which has been used for flexibility (Ribeiro et al., 2016a). However, this can lead people to misunderstanding the explanation because it is not offered by the model that provides the prediction. Therefore, as Rudin (2019) claimed, it is necessary to use an inherently-interpretable model in order to accurately explain DNNs without any mistakes, which results from an architecture to provide intrinsic explanation.

We build an inherently interpretable model, by referring to methods that generate prototypes and conduct classification using Euclidean-distance in a latent space (Li et al., 2018; Chen et al., 2019). These prototype-based explanation methods provide explanations with examples and distances and show efficacy in diverse domains. However, prototype-based methods are powerful but not enough to understand a model prediction fully, as explanations must be both human-interpretable and reliable. As most of the existing prototype-based explanation models suffer from the problem that similar latent representations do not guarantee similarity in shape, and distances often provide explanations that are not human-interpretable. Users can raise doubts, but have no choice but to accept the given distances only as numerical values. Since there is no way to verify how distances are constructed, the reliability of the explanations is compromised.

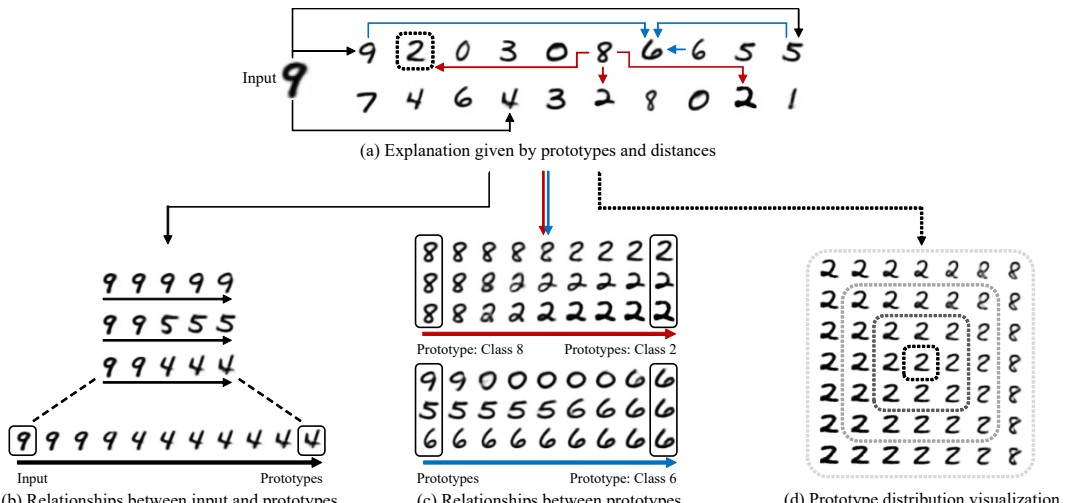

Figure 1: Basic explanation: (a) Learned prototypes that represent the training dataset and distances between the input and prototypes. Additional explanations(b-d): (b) Relationships between input and prototypes. Bottom of the figure explains that the explanation space enables us to see how input 9 gradually becomes prototype 4. (c) Relationships between prototypes. Providing gradual changes from prototype to another prototype that explains how model discriminates prototypes. (d) Prototype distribution visualization with two dimensions based on standard deviation demonstrating that prototypes are not luckily chosen.

To solve these problems, we construct an *explanation space* that provides gradual explanations and reliable distance. By using VAE, a latent space can be visualized as an image that users can recognize. We also use VAE training techniques to regulate a latent space to ensure latent representations share similarities in appearance. This means that a distance in the latent space can have more "intrinsic meaning". We call the latent space obtained through the process the explanation space. Our model provides additional gradual explanation as in Figure 1. Meanwhile, the prototypes and the distances between the input data and prototypes are given as basic explanation in Figure 1(a). Through these relationships, users are able to understand how distances are composed. Instead of providing the distances between the input and prototypes, it is possible to better understand a model by showing which features are changing in terms of relationship. Therefore, our proposed model allows users to interpret distances more reliably.

In this paper, we developed the model that performs classification through distances between prototype distributions and the encoded input distribution in the latent space of the Variational Autoencoder (VAE)(Kingma & Welling, 2014). After VAE encodes input data in a Gaussian distribution form, we compare the encoded input distribution and generated prototype distributions through four measures with various number of prototype distributions to provide explanations with high classification accuracy. Furthermore, we use the method (Sensoy et al., 2018) of placing a Dirichlet distribution on the class probabilities to improve accuracy by reducing the uncertainty of the model.

**Contributions.** Our key contributions are as follows:

- By constructing the explanation space, we can visualize additional gradual explanations. Through gradual explanations, relationships are provided rather than just providing each prototype image.

- By our dense explanation space approach using VAE, similar latent representations tend to resemble each other in appearance. Therefore, distance as explanation is more meaningful because similarity is guaranteed.

- We present a method to find a baseline number of prototypes to describe the training dataset. In experiments, we show changes in accuracy and explanations of the model by varying numbers of prototypes.

## 2 RELATED WORK

**Explanation Methods for DNN** While there are diverse suites of methods to explain DNN, it can be categorized based on the access of the model's internal information and how they provide explanations (Jeyakumar et al., 2020). The model-agnostic method is to approximate the relationships between input data and output decision while treating the model as a black-box. There are methods to create a local explanation (Ribeiro et al., 2016b; Chen et al., 2018a; Lundberg & Lee, 2017; Chen et al., 2018b) by selecting features that play an essential role in making decisions and methods providing an interpretable model through a global approximate (Guo et al., 2018). On the other hand, in knowing the internal information of the model, the model-transparent method provides explanations by saliency map using gradient in classifier (Simonyan et al., 2014; Smilkov et al., 2017; Sundararajan et al., 2017), class activation map using global average pooling (Zhou et al., 2016; Selvaraju et al., 2017), and feature map based on attention (Liu et al., 2020). The third method provides explanations by examples capturing relationship between test input and the model's decision. Considering that examples are based on training data (Bien & Tibshirani, 2011; Koh & Liang, 2017; Arık & Pfister, 2020) or learned prototype data (Li et al., 2018; Chen et al., 2019; Hase et al., 2019), our work is closely aligned with these prototype-based classification models. The example-based method shows its efficacy to explain complex concepts in psychology (Aamodt & Plaza, 1994) and education (Renkl, 2014) domains, and is also accepted by the general public as the most preferred explanation method to explain DNN (Jeyakumar et al., 2020).

The following three prototype-based classification methods are similar to our method in that they use the distance between the encoded input and prototypes for classification and explanation. Li et al. (2018) learn and visualize the prototypes for the entire data through autoencoder. Chen et al. (2019) go further from here, characterizing training image patches as class-specific prototypes for partial explanations, while Hase et al. (2019) feature a hierarchical demonstration of prototypes through related groups for non-flat explanations. As most of prototype-based explanation methods are based on Li et al. (2018), we use it as a baseline model. Our framework goes further from these methods, expands prototypes from vectors to distribution in latent space and constructs explanation space for diverse explanations with reliability. In addition to prototype-based methods, there are several methods that use the distance in latent space to understand the internal behavior of a model or to provide explanation using VAE (Antorán et al., 2021; O'Shaughnessy et al., 2020) or GAN (Lang et al., 2021; Yang et al., 2021).

**Uncertainty in deep learning** The largest body of research on estimating uncertainty in deep learning is Bayesian neural networks (Neal, 2012; Gal & Ghahramani, 2015; Kendall & Gal, 2017). To obtain predictive uncertainty, Bayesian neural networks use prior distributions on model parameters and infer the posterior distribution. Although exact inference in Bayesian methods is intractable, a range of approximate techniques has been proposed, such as variational inference (Kingma et al., 2015; Gal & Ghahramani, 2016) and stochastic gradient hamiltonian monte carlo (SG-HMC) (Chen et al., 2014). However, in practice, these Bayesian methods are outperformed by Deep Ensembles (Lakshminarayanan et al., 2017). This non-bayesian technique uses multiple neural networks trained from different initializations and averages their predictions as the model output. It is simple and effective but requires high computational costs at training and test time. Another non-bayesian technique is based on evidential deep learning (Sensoy et al., 2018; Amini et al., 2019), which we use to estimate uncertainty. These methods use a single deterministic neural network and apply prior distributions on model predictions, not model parameters.

## 3 METHODS

Our model consists of three parts: (i) VAE layer, (ii) Distance measuring layer (iii) Uncertainty layer, as shown in Figure 2. The VAE layer is used to encode input into a form of distribution and decode the values obtained at prototype distributions and relationships with recognizable image data. The distance between prototype distributions and encoded input distribution is used as input for one fully-connected layer to create an inherently interpretable classification model. Moreover, since classification is performed with distances in the latent space of VAE, we use an uncertainty layer to increase accuracy.

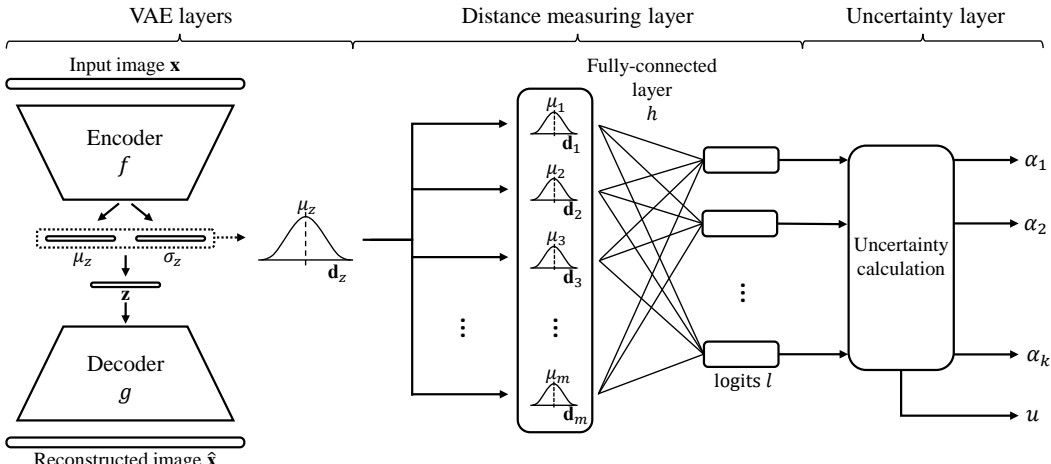

Figure 2: Overall model architecture. The entire model is trained end-to-end. Since the uncertainty layer was used to reduce uncertainty in the training process, classification proceeds through logits. The explanation is provided by showing the learned prototypes, the distance between the prototypes and $\mu_z$, and their relationship to each other through the latent space.

## 3.1 USING VARIATIONAL AUTO ENCODER

Let $\mathcal{D} = \{X, Y\}$ be the training dataset consists of images $\mathbf{x} \in \mathbb{R}^N$ and labels $\mathbf{y} \in \{1, \ldots, K\}$. The encoder $f : q(\mathbf{z}|\mathbf{x})$ of the VAE maps an input image to a Gaussian distribution $\mathbf{d}_z \sim \mathcal{N}(\mu_z, \sigma_z{}^2)(\mu_z, \sigma_z \in \mathbb{R}^d)$ which is reparameterized to generate latent vector $\mathbf{z} \in \mathbb{R}^d$. After that, the generated $\mathbf{z}$ is reconstructed in the form of image $\hat{\mathbf{x}} = g(\mathbf{z})$ through decoder $g : p(\mathbf{x}|\mathbf{z})$. Unlike autoencoder, VAE regulates the latent space created by the encoder as isotropic unit Gaussian ($p(\mathbf{z}) \sim \mathcal{N}(0, I)$), which is why we used VAE as a feature extractor. By regularizing latent space and sampling latent vectors from distribution, VAE constructs appropriate dense latent space that creates similar latent representations from similar-shaped images, which plays an essential role in the interpretability of our method described in Section 4.

Furthermore, our model learns prototype distributions as $m$ independent multivariate Gaussian distributions $\mathbf{D} = \{\mathbf{d}_1, \mathbf{d}_2, \ldots, \mathbf{d}_m\}, \mathbf{d}_i \sim \mathcal{N}(\mu_i, \sigma_i^2) \in \mathbb{R}^d$ and visualizes $\mu_i$ as prototype. For that, the structures of disentangled VAE are applied in order to disentangle the explanation components at prototype distributions through dimensions. A method of encouraging the independence of representation distribution $q(\mathbf{z})$ by lowering the KL divergence between $q(\mathbf{z})$ and $\bar{q}(\mathbf{z}) := \prod_{j=1}^{d} q(\mathbf{z}_j)$, and methods of lowering the KL divergence of encoder $f$ and isotropic unit Gaussian $p(\mathbf{z})$ to constrain the latent space stronger are additionally used with vanilla VAE. Applying these methods would result in the following objective:

$$\frac{1}{n} \sum_{i=1}^{n} \left[ \mathbb{E}_{q(\mathbf{z}|\mathbf{x}_i)}[\log p(\mathbf{x}_i|\mathbf{z})] - \beta KL(q(\mathbf{z}|\mathbf{x}_i)\|p(\mathbf{z})) \right] - \gamma KL(q(\mathbf{z})\|\bar{q}(\mathbf{z})).$$

In this objective function, $\beta$ and $\gamma$ are hyperparameters that control the power of the constraint terms.

## 3.2 CLASSIFICATION NETWORK USING DISTANCE

Once the input is encoded by latent distribution $\mathbf{d}_z$, logits based on the distance from the prototype distributions are produced for the classification. This process is divided into (i) measuring the distance between learned $\mathbf{D} = \{\mathbf{d}_1, \mathbf{d}_2, \ldots, \mathbf{d}_m\}$ and $\mathbf{d}_z$ in the latent space and (ii) producing the logits by measured distance using the fully-connected layer $h$ for the classification.

Since prototype distributions $\mathbf{D}$ and encoded input $\mathbf{d}_z$ both are in the form of Gaussian distribution, we use 2-Wasserstein distance (Givens et al., 1984) as a distance measure. The 2-Wasserstein distance $W$ for two multivariate Gaussian distribution $\mathcal{N}(\mu_1, \Sigma_1), \mathcal{N}(\mu_2, \Sigma_2)(\mu_i \in \mathbb{R}^d, \Sigma_i \in \mathbb{R}^d \times \mathbb{R}^d)$ can

be computed as:

$$W_{12}^2 = \|\mu_1 - \mu_2\|_2^2 + Tr(\Sigma_1 + \Sigma_2 - 2(\Sigma_1^{1/2}\Sigma_2\Sigma_1^{1/2})^{1/2}).$$

Since we assume the prototype distributions $\mathbf{D}$ to be independent and encoder $f$ predicts diagonal covariance matrices, $\Sigma_i = \mathrm{diag}(\sigma_i^1, \ldots, \sigma_i^d)$ and the above equation simplified as:

$$W_{12}^2 = \sum_{j=1}^d [(\mu_1^j - \mu_2^j)^2 + (\sigma_1^j - \sigma_2^j)^2].$$

Three other measurements KL-divergence (KLD), Jensen-Shannon divergence (JSD), Jensen-Tsallis distance (JTD) (Choi et al., 2019) are also experimented in Section 5.1.

The distances between $\mathbf{D}$ and $\mathbf{d}_z$ are used as input of fully-connected layer $h$ to generate logits. The reason for classification by passing only one fully-connected layer is to construct a weight matrix. By constructing a weight matrix using $h$, the model can numerically confirm the impact of each prototype distribution on class selection. This weight matrix gives interpretability to the model's prediction even when the number of given prototype distributions $m$ and the class $K$ are different.

### 3.3 USING UNCERTAINTY LAYER

To repeat sampling could increase the uncertainty of classification based on distances in a latent space. Therefore, by considering uncertainty, we also improve the classification performance by about 1% with the method (Sensoy et al., 2018) that uses the Dirichlet distribution to reduce the uncertainty of a model. In the training process, the uncertainty loss $\mathcal{L}_{\text{uncertain}}$ is calculated using the Dirichlet parameters obtained through logits $l$. The detailed method for uncertainty layer is in Appendix A.5.

After going through all of these processes, the final objective function is composed as follows:

$$\mathcal{L}(\mathbf{x}, \mathbf{y}) = \lambda \mathcal{L}_{\text{uncertain}}(h(W(\mathbf{D}, f(\mathbf{x}))), \mathbf{y}) + \mathcal{L}_{\text{VAE}}(\mathbf{x}, \hat{\mathbf{x}}),$$

where $\mathcal{L}_{\text{uncertain}}$ is the uncertainty classification term based on cross-entropy loss and $\mathbf{D}$ and $\mathbf{d}_z$ is the VAE term that described at Section 3.1. Moreover, $\lambda$ is a hyperparameter for adjusting the ratio between them.

## 4 CONSTRUCTING INTERPRETABLE EXPLANATIONS

### 4.1 RELIABLE EXPLANATION SPACE

Our method increases the reliability of the explanation through distance and the prototypes. As noted in Brendel & Bethge (2019), the interpretation of previous prototype-based classification methods is difficult because only the Euclidean distance between prototypes and latent representations is given, while having a similar latent representation does not guarantee that images share similarities in human-interpretable features. Therefore, by sampling with distributions and regularizing latent space through VAE, our model constructs dense explanation space that ensures similar latent representations share similarities in appearance. Figure 3 makes this more evident. Looking at the visualized latent space by t-SNE (Van der Maaten & Hinton, 2008), embedded data by the previous method tends to separate clusters, often make the case that the distance between two points in the same class farther than the distance between two different class points, as shown in Figure 3(a). In contrast, the proposed model increases the reliability of explanation through distance by showing that each class is clustered together and exist close to each other that forms dense latent space. Furthermore, by showing latent representations contained within the prototype distribution $\mathbf{d}_i$ as images, the prototype's reliability can also be increased by their similarities. As illustrated in Figure 1(d), features within the distribution are similar to the prototype $\mu_i$ at basic explanation, demonstrating that prototypes are not coincidentally chosen.

We made quantitative comparisons through user study as well as qualitative comparisons to ensure the reliability of the explanation space. We measured the user's understanding through an interactive reconstruction experiment (Ross et al., 2021) to show the reliability of the explanation space of our method. This experiment is to generate an image similar to a randomly sampled image by changing

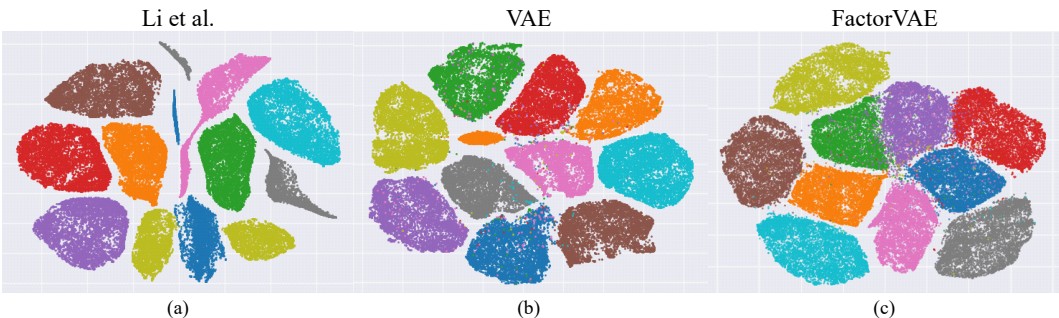

Figure 3: Visualizations for the latent space of FashionMNIST by three models. (a) Li et al. (2018) (b) Proposed model with VAE. (c) Proposed model with FactorVAE.

each dimension value of the latent space through a slide using the decoder of the models. As the user explores the latent space and generates an image similar to the sampled image, we can see how the configuration of the latent space is easy to understand for the user. The evaluation was conducted through Amazon Mechanical Turk. In order to improve the quality of the evaluation, all experiments were conducted through 20 Mechanical Turk Masters. The success rate by model is shown in Table 1. Detailed settings for experiments are in Appendix A.2.

As Ross et al. (2021) concluded that a higher success rate on interactive reconstruction tasks shows improved interpretability, we concluded that the explanation space by our method improved interpretability because the success rate obtained through the interactive reconstruction experiment showed a higher value than the baseline model. The distance value in the explanation space having more intrinsic meaning because the explanation space created by our method showed improved interpretability compared to the baseline model. Therefore, the experimental results could be used as quantitative evaluation to support that our method created a reliable explanation space.

Table 1: Success rate by baseline model (Li et al., 2018), proposed model with VAE, FactorVAE

|  | Success rate |
|---|---|
| Li et al. | 0.27±0.28 |
| VAE | 0.48±0.31 |
| FactorVAE | 0.54±0.29 |

## 4.2 SELECTING THE NUMBER OF PROTOTYPE DISTRIBUTIONS

In the prototype-based explanation model, the number of prototypes is critical. As can be seen in the model structure, the number of prototypes affects the input of the fully-connected layer and inevitably affects accuracy. Moreover, since explanations are also provided through prototypes, it is important to set the number of prototypes. However, since there was no information to refer other than the number of classes to determine the number of prototypes that can represent the dataset, the previous methods (Li et al., 2018; Chen et al., 2019; Hase et al., 2019) specified the number of prototypes to an arbitrary and heuristic value determined by a human.

We propose a method to produce the baseline number of prototype distributions in our framework. Since we have set the prototype distributions as independent Gaussian distributions rather than vectors, prototype distributions cover the latent space. Furthermore, as prototype distributions are representative of the whole dataset, it shares similarities with the Gaussian Mixture Model (GMM) in that the whole dataset is represented in the form of multiple Gaussian distributions. We choose an optimal number of components of the GMM in latent space as a baseline number of prototype distributions. We use the Bayesian Information Criterion (BIC), which is known to be the most appropriate criterion for GMM (Steele & Raftery, 2010). The experimental results for this can be seen in the Section 5.3.

## 4.3 EXPLAINING THE RELATIONSHIP OF EXPLANATIONS

The explanations of our model are divided into basic explanation and explanations of the basic explanation. For fundamental explanations, visualized images of the learned prototypes and the distance between encoded input and prototype distributions are given as previous prototype-based explanation methods. To explain this basic explanation, we additionally provide two gradual explanations. These

explanations can be obtained because our method constructs dense explanation space, unlike previous methods. As a result, it is possible to generate interconnection between latent representations as images. We represent changing elements by interpolating these latent representations and define them as relationships.

The relationships between inputs and prototypes can be visualized by our model. In situations where distance is given for interpretable information, the factors that can reduce the distance are critical for understanding the model's prediction. Additionally, relationships between prototypes are given. Since the prototypes are representative of the whole dataset, relationships between prototypes show which features are needed to change from one class to another class. This explanation can be used as basic information about how the model discriminates classes.

After the model is trained, basic explanation and explanations for basic explanation are visualized as follows. First, the prototype, provided as the fundamental explanation, is visualized by sending $\mu_i$ of the prototype distribution through the decoder $g$ as a prototype. Next, the relationships between prototype $\mu_i$ and prototype $\mu_j$ are visualized as sending $\mu_i + (\mu_j - \mu_i)r/k$ for $r = 0, 1, \ldots, k$ through the decoder $g$, which can be controlled by increasing the value of $k$ for detailed explanation and decreasing the value of $k$ for a brief explanation. Finally, the relationships between inputs and prototypes are provided in the same way, and visualizations of prototype distributions are based on $\mu$ values and $\sigma$ values.

The disadvantage of our model comes from this visualization. Many training samples are required to generate explanation images. In addition, visualized images of our explanations are fundamentally generated through VAE, making it challenging to generate human-recognizable images for complex datasets that are difficult for VAE to generate the observable image. However, this is a problem related to VAE's expressive power, and there are many high-resolution expressive methods (Lee et al., 2020; Vahdat & Kautz, 2020) based on VAE, so applying these methods would solve these problems in future work.

## 5 EXPERIMENTS

**Experiment details** MNIST and Fashion-MNIST (Xiao et al., 2017) datasets are used for our experiments and are splited by 0.9, 0.1 for training dataset and test dataset. For training our model, we use Adam (Kingma & Ba, 2015) optimizer with learning rate of 0.0001 and weight decay of 0.00005, and set hyperparameter $\lambda$ to 50. To compare various VAE structures, we use vanilla VAE ($\beta = 1, \gamma = 0$), $\beta$-VAE ($\beta = 4, \gamma = 0$) (Higgins et al., 2017) and FactorVAE ($\beta = 1, \gamma = 6.4$) (Kim & Mnih, 2018). We train 500 epochs with VAE and $\beta$-VAE for about 2 hours, and about 3 hours were used with FactorVAE using a single 1080 Ti GPU.

Table 2: Accuracy by measures(%) on Fashion-MNIST

| Measure | VAE | | Beta-VAE | | FactorVAE | |
|---|---|---|---|---|---|---|
| | original | uncertain | original | uncertain | original | uncertain |
| KLD | 91.3±0.2 | 92.1±0.2 | 91.6±0.3 | 92.2±0.3 | 91.2±0.2 | 92.4±0.1 |
| JSD | 91.9±0.2 | 92.4±0.2 | 91.9±0.2 | 92.4±0.3 | 91.9±0.3 | 92.4±0.2 |
| JTD | 91.0±0.1 | 92.1±0.2 | 91.4±0.2 | 92.1±0.2 | 92.1±0.2 | 92.0±0.3 |
| WSD | 91.8±0.1 | **93.0 ± 0.1** | 92.2±0.2 | **92.6 ± 0.2** | 91.3±0.2 | **92.8 ± 0.1** |

### 5.1 CLASSIFICATION ACCURACY

The accuracy of our model on MNIST is 99.33±0.04% with providing visualized explanations in Figure 1. Furthermore, as shown in Table 2, our model achieved testing accuracy 93.00±0.13% on Fashion-MNIST. This result is comparable to non-interpretable classification networks such as VGG16 (93.5%) and GoogleNet (93.7%) reported in the dataset website. Moreover, we achieve higher accuracy than the baseline model's (Li et al., 2018) accuracy 91.03±0.32% that only provides basic explanation. We construct baseline model with same layers and same hyperparameters.

As shown in Table 2, we conduct experiments with four measurements and three VAE structures. According to measures, results of experiments using Wasserstein-distance show the highest accuracy

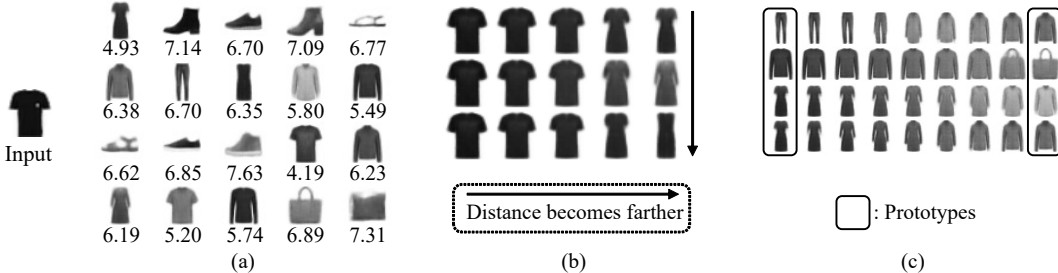

Figure 4: Explanations of classification network with 20 prototype distributions and FactorVAE. (a) Prototypes and distance between input and prototype distributions in the latent space. (b) Relationships between input and prototypes. (c) Relationships between prototypes.

through three VAE structures. In addition, it achieves the highest diversity and stability of the learned prototypes as it provides a usable gradient when distributions are supported on non-overlapping domains (Kolouri et al., 2019). This indicates that Wasserstein-distance is the most suitable measurement in creating prototypes using distances between distributions. The role of the uncertainty layer to increase accuracy is also clearly revealed. In most of the 12 cases, accuracy increased when the uncertainty layer is applied. Among three VAE structures, the vanilla VAE shows the highest accuracy, but FactorVAE also shows no significant difference in accuracy. However, FactorVAE forms more reliable explanation space as Section 4.1, it is used to create Figures 1, 4, 5.

## 5.2 PROVIDED EXPLANATIONS: BEYOND EXAMPLES

Our model generates explanations for prediction as Figure 4. In Figure 4(a), for the black T-shirt input image, our model shows prototypes with the lowest distance having shapes of tops. On the other hand, the distances from the prototypes that looked like bags, trousers, and shoes are higher than 6.5, indicating that the prototypes and distance information provide accurate explanations. Furthermore, a fully-connected layer is applied as a weight matrix corresponding to a similarity connection between these distances and logits, showing relevance between prototype distributions and classes. However, this explanation alone makes it difficult to understand why the distance from the prototypes in the same T-shirt class is different and whether the different distances came out even if the prototypes were in the form of dresses. Therefore, our method provides additional explanations for this basic explanation.

Additionally, our model provides the relationships between prototypes and inputs as an explanation of the factors that influenced distance, such as Figure 4(b). For example, from relationships between the input and prototypes for class dress, the nearest prototype can be seen with only the thickness of the torso changing, followed by the next prototype with longer sleeves and the farthest prototype with no sleeves. Since prototypes with changes in the sleeves have similar distances, we can assume that the model is sensitive to sleeves in the relationship between input and dress prototypes. The bigger the change in the sleeves of the clothes, the farther the distance becomes. In addition, relationships between prototypes are visualized as additional information for how the model discriminates prototypes, such as Figure 4(c).

We conduct a user study to show that the relationship helps us to understand model predictions. With the explanations from our model, subjective comparison is made when (i) basic explanation (Figure 4(a)), and (ii) basic explanation and the relationship between input and prototype (Figure 4(a), (b)) are given. For each set of explanations, we studied the results of correct/wrong classification. We prepare the following questions to check whether users were able to respond to factors that influenced model predictions. In the case of correct classification, we ask the factors that affect differences in the distance between prototypes within the same class. In the case of misclassification, we asked the factors that made the distance from the prototype of the wrong class close. The relationships with the input and following prototypes were given during the survey: (i) Correct classification: two prototypes of the same class as input. (ii) Wrong classification: the prototype closest to the input among misclassified classes, and the prototype closest to the input among true classes. Additional information about the survey is included in Appendix A.3.

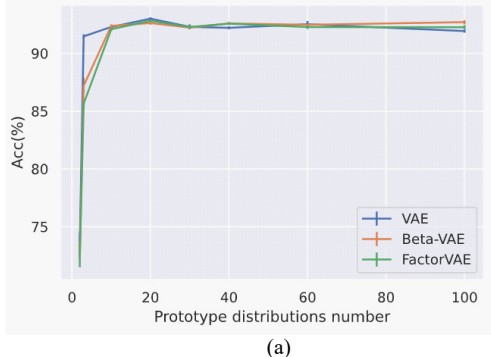 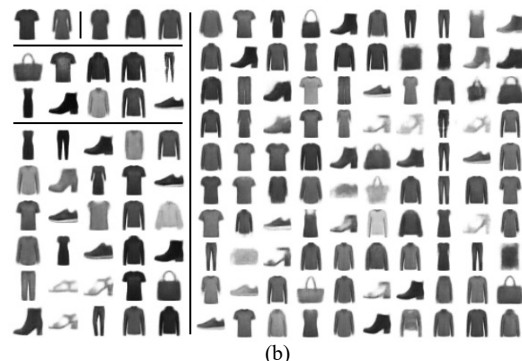

(a)                                          (b)

Figure 5: Accuracy and visualizations of prototypes. (a) Classification accuracy by changing numbers of prototype distributions (b) Visualization of prototypes $\mu_i (i = 1, \ldots, m)$ at number of prototype distributions $m = [2, 3, 10, 30, 100]$

We conducted two surveys through Amazon Mechanical Turk with 50 Mechanical Turk masters. Each survey consists of 12 questions, 6 per correct classification and misclassification. The average response rate for answering the factor is in Table 3. This result shows that participants were better able to answer factors for model predictions when given the additional explanations of the relationship between inputs and prototypes. It shows that relationships enables us to better understand the factors that compute distances.

Table 3: Response rate without/with relationships

|  | Without | With |
|---|---|---|
| Correct cls. | 0.77 | 0.83 |
| Wrong cls. | 0.60 | 0.71 |

### 5.3 CHANGING THE NUMBER OF PROTOTYPE DISTRIBUTIONS

As mentioned above, we chose the optimal number of Gaussian Mixture Model's components as a proper number of prototype distributions. We use the EM algorithm to fit Gaussian Mixture Model to the dataset on latent space. First, trained VAE encodes $\mu$ values of the entire dataset to place them on the latent space. Since our prototype distributions consist of independent multivariate Gaussian distributions, we use the Gaussian Mixture Model with each component having its own diagonal covariance matrix. As the first local minimum BIC score indicates strong evidence for the model (Fraley & Raftery, 1998), we determine that the number of components at the first local minimum is the optimal number of components. The number of components obtained is 20.3, 19.8, 23.8 on average in VAE, $\beta$-VAE, and FactorVAE. Through these results, the baseline number of prototype distributions is set as 20. To verify this, we experiment with numbers of prototype distributions $m = [2, 3, 10, 20, 30, 40, 60, 100]$ and the results are in Figure 5. In Figure 5(a), the accuracy begins to converge around 20. In addition, when the number of prototypes are small, the model converges extensively slowly. In Figure 5(b), when the number of prototypes increases from a number smaller than the number of dataset classes, various appearances appear. However, when the number of prototypes increases excessively, similar appearance prototypes are inevitable. These results indicate that it is better to set the number of Gaussian mixture model components as the number of prototype distributions.

## 6 CONCLUSION

The main contribution of our paper is to further enhance human understanding of model predictions by providing additional explanations of relationships. Further to existing prototype-based explanation methods that generate explanations by prototype examples and distances, our model visualizes relationships between input and prototypes that can explain the factors relevant to distance. Moreover, due to creating a dense explanation space, the distance became possible to function as a numerical value that guarantees similarity, which led to increase reliability.

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

# A APPENDIX

## A.1 MODEL ARCHITECTURE DETAILS

In our proposed model, we use vanilla VAE, $\beta$-VAE, and FactorVAE for the VAE layer. For VAE and $\beta$-VAE, we use a Convolutional Neural network for the encoder and a Deconvolutional Neural Network for the decoder. Detailed architecture explanation is provided in Table 4. For FactorVAE, discriminator is additionally used for distinguishing whether the input was drawn from $q(\mathbf{z})$ or $\bar{q}(\mathbf{z}) := \prod_{j=1}^{d} q(\mathbf{z}_j)$ to calculate KL divergence between $q(\mathbf{z})$ and $\bar{q}(\mathbf{z})$. The discriminator is made of 6 MLP layers with 1000 hidden units each and leaky ReLU is used for non-linearity, that generates 2 logits as outputs. We use Adam optimizer for discriminator with learning rate 0.0001 and weight decay 0.00005.

## A.2 DETAILS FOR INTERACTIVE RECONSTRUCTION EXPERIMENT

We constructed the experiments in the same way as in the paper Ross et al. (2021). Through the interactive reconstruction method shown in the paper, we evaluate the models: Li et al. (2018), proposed model with VAE, and proposed model with FactorVAE. The evaluation was conducted through Amazon Mechanical Turk, and for a fair evaluation, everyone paid $3.75 per evaluation. In order to improve the quality of the evaluation, all experiments were conducted through Mechanical Turk Masters.

The experiments were conducted with models trained through the MNIST dataset. Success was considered if the user-generated image and the sampled image had a similarity of at least 75%. In addition, it is designed so that users can skip the question only if they generate images for at least 45 seconds. Experiments were conducted with 20 participants per model and 7 questions with random images were provided. An example of interactive reconstruction experiment is shown in Figure 6.

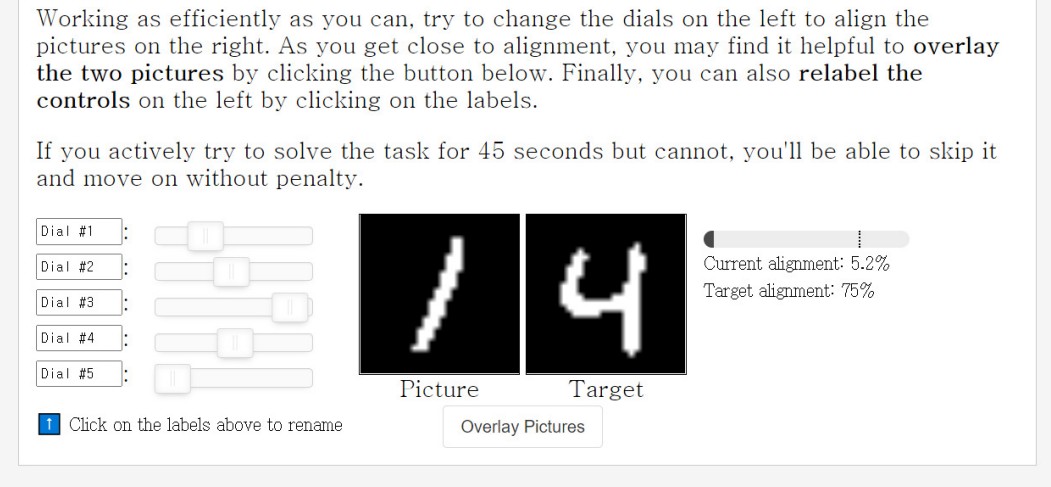

Figure 6: Example of interactive reconstruction experiment.

### A.3 User study about relationships

The survey was conducted after showing precautions to the participants first. Participants were asked to answer the factors that influenced the model's decision by looking at the given information. At this time, since the given information may be insufficient to judge the factor, two options were provided: the option of not knowing and the option of guessing. In the case of the participants who judged that they could guess factors, they were also asked to answer which factors they guessed. Consent form and example of survey without relationships is shown in Figure 7. Consent form and example of survey with relationships is shown in Figure 8.

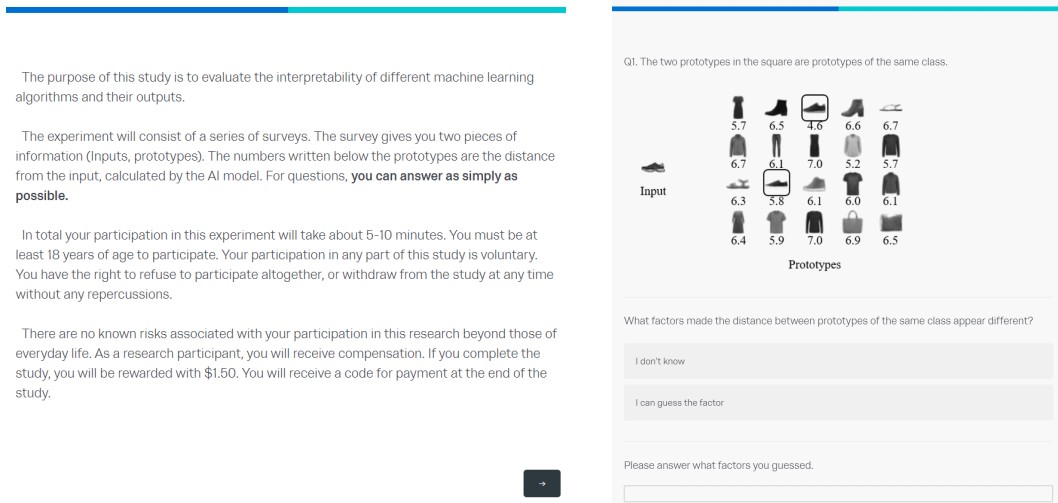

Figure 7: Consent form and example of survey without relationships.

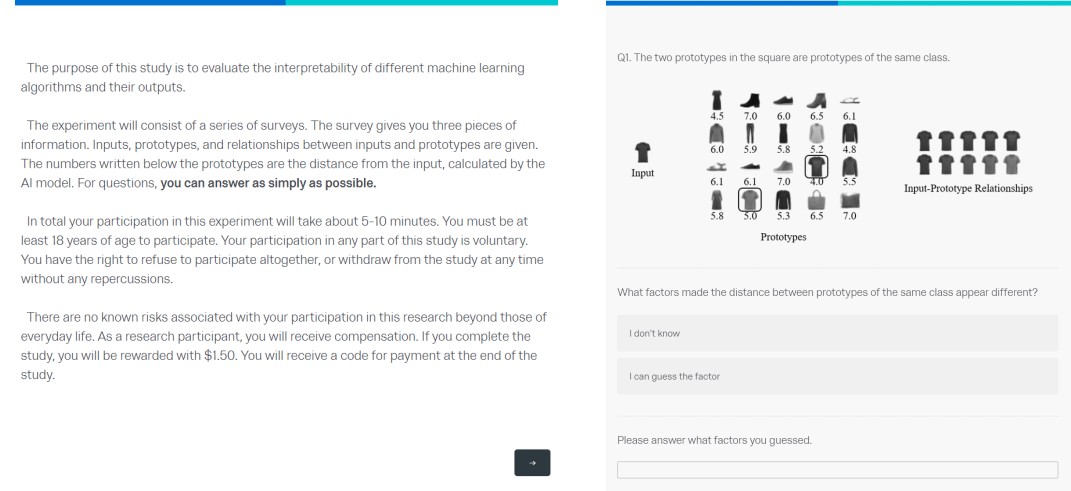

Figure 8: Consent form and example of survey with relationships.

### A.4 Detail explanations for measurements

We use KL-divergence, Jensen-Shannon divergence (JSD), Jensen-Tsallis distance (JTD) and Wasserstein distance for measuring distance between prototype distributions $\mathbf{D} = \{\mathbf{d}_1, \mathbf{d}_2, \ldots, \mathbf{d}_m\}$ and encoded input distribution $\mathbf{d}_z$.

Since it was not described in detail in this paper, we provide additional explanations in this section. For two multivariate Gaussian distributions $p = \mathcal{N}(\mu_1, \Sigma_1), q = \mathcal{N}(\mu_2, \Sigma_2) (\mu_i \in \mathbb{R}^d, \Sigma_i \in \mathbb{R}^d \times \mathbb{R}^d)$, KL-divergence is as follows:

$$KL(p||q) = \frac{1}{2}\left[\log\frac{|\Sigma_2|}{|\Sigma_1|} - d + tr\left\{\Sigma_2^{-1}\Sigma_1\right\} + (\mu_2 - \mu_1)^T\Sigma_2^{-1}(\mu_2 - \mu_1)\right]. \quad (1)$$

From independent distributions $\mathbf{D}$ and encoded input distribution that has diagonal covariance matrix $\Sigma_i = \mathrm{diag}(\sigma_i^1, \ldots, \sigma_i^d)$, the above equation simplifies to:

$$\sum_{j=1}^{d}\left[\log\frac{\sigma_2^j}{\sigma_1^j} + \frac{(\sigma_1^j)^2 + (\mu_1^j - \mu_2^j)^2}{2(\sigma_2^j)^2} - \frac{1}{2}\right]. \quad (2)$$

Through this equation, with $m = \frac{1}{2}(p + q)$, Jensen-Shannon divergence also simplifies in the same manner because Jensen-Shannon divergence defined as:

$$JSD(p||q) = \frac{1}{2}(KL(p||m) + KL(q||m)). \quad (3)$$

Unlike JSD, Jensen-Tsallis divergence uses the Tsallis entropy instead of the Shannon entropy. The Tsallis entropy $S_{q,k}(p)$ is generalization of the Shannon entropy which becomes Shannon entropy when $q$ converges to 1 and $k$ equals 1. In previous works, they use specific Jensen-Tsallis divergence with $q = 2$ and $k = 4$ and refer to JTD since it satisfies the conditions of a metric. In our condition, JTD is as follows:

$$JTD^2(p||q) = \sum_{j=1}^{d}\left[\mathcal{N}(\mu_1^j; \mu_1^j, 2(\sigma_1^j)^2) + \mathcal{N}(\mu_2^j; \mu_2^j, 2(\sigma_2^j)^2) - \mathcal{N}(\mu_1^j; \mu_2^j, (\sigma_1^j)^2 + (\sigma_2^j)^2)\right]. \quad (4)$$

By using KL-divergence, JSD, and JTD as a distance measure, the proposed model learns prototypes as illustrated in Figure 9. Using these metric for distance, the model accuracy is high enough, but the prototypes do not form observable images, which is contrary to our purpose of explaining the model predictions. This is because these three measures fail to provide usable gradients when distributions are supported on non-overlapping domains. For this reason, the model fails to provide usable gradients when the prototype distributions and the embedded input distribution do not overlap, thus not forming a proper latent space as shown in the Figure 9, which makes prototypes unstable.

## A.5 DETAILED ANALYSIS DUE TO APPLICATION OF UNCERTAINTY LAYER

In the process of forming the latent space in our proposed model, VAE formulates the reconstruction loss through the latent vector sampled from the embedded input distribution. Since classification was performed through the distance in the latent space, we tried to improve the classification accuracy by lowering the uncertainty of the model. By adding the calculation process to logits without changing the structure of the model, it was possible to predict uncertainty and increase the accuracy by making the model robust against adversarial perturbations. Furthermore, since the model is not trained to handle uncertain information, it not only increases accuracy but also makes similar features exist in similar positions in the latent space. The embedded dataset visualized with t-SNE is illustrated in Figure 10 showing that data of the same class forms more aggregated clusters in latent space when the uncertainty layer is applied. Through this latent space property, prototypes form more obvious observations that can express the whole dataset better, as Figure 11, because prototype distributions are learned through distances at latent space.

For detailed explanations of the uncertainty layer setting in our model is as follows. Given a sample $\mathbf{x}_i$, let there be one-hot vector $y_i$ with ground truth class $j$, then $y_{ij} = 1$ and $y_{ik} = 0$ for all $k \neq j$, and logit $l_i$ that comes through the distance measuring layer of the model. We compute the evidence $e_i$ as exponential of logit $l_i$. Then use the evidence to derive the uncertainty probability $u_i = K/S_i$, where total $S_i = \sum_{j=1}^{K}(e_{ij} + 1)$, and parameters of Dirichlet distribution $\alpha_{ik} = e_{ik} + 1$ for class

Table 4: VAE network architecture used for experiments with MNIST and FashionMNIST

| VAE Encoder | VAE Decoder |
|---|---|
| Input 32×32 greyscale image | Input z $\in \mathbb{R}^{10}$ |
| Conv2d (32 channels, 4×4 kernels, stride 2, pad 1) | Conv2d (256 channels, 1×1 kernels, stride 1, pad 0) |
| BatchNorm, LeakyReLU(0.2) | BatchNorm, LeakyReLU(0.2) |
| Conv2d (32 channels, 4×4 kernels, stride 2, pad 1) | ConvTranspose2d (64 channels, 4×4 kernels, stride 1, pad 0) |
| BatchNorm, LeakyReLU(0.2) | BatchNorm, LeakyReLU(0.2) |
| Conv2d (64 channels, 4×4 kernels, stride 2, pad 1) | ConvTranspose2d (32 channels, 4×4 kernels, stride 2, pad 1) |
| BatchNorm, LeakyReLU(0.2) | BatchNorm, LeakyReLU(0.2) |
| Conv2d (256 channels, 4×4 kernels, stride 1, pad 0) | ConvTranspose2d (32 channels, 4×4 kernels, stride 2, pad 1) |
| BatchNorm, LeakyReLU(0.2) | BatchNorm, LeakyReLU(0.2) |
| Conv2d (2*10 channels, 1×1 kernels, stride 1, pad 0) | ConvTranspose2d (1 channels, 4×4 kernels, stride 2, pad 1) |
|  | Sigmoid |

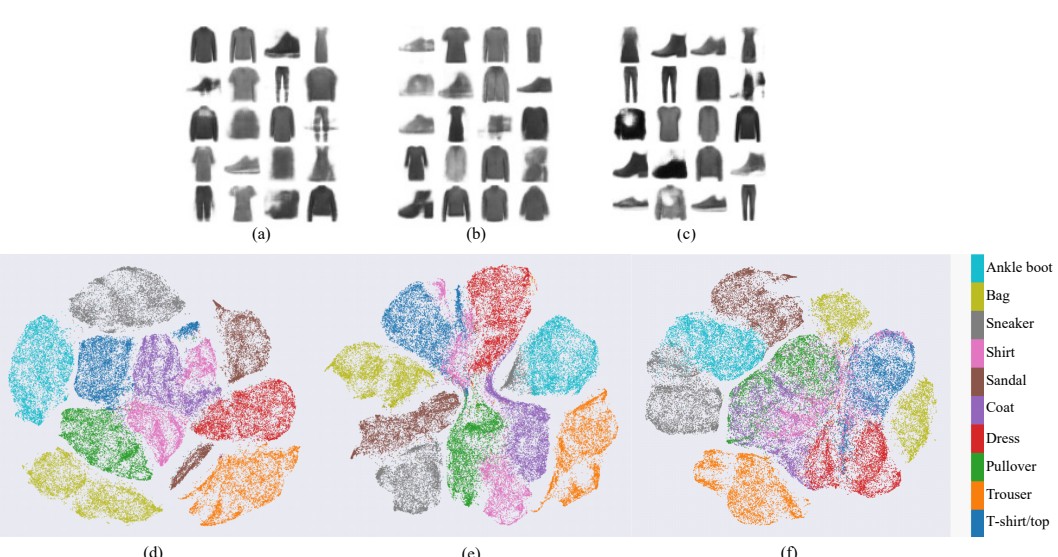

Figure 9: Prototypes made with the proposed model using (a) KL-divergence, (b) JSD, and (c) JTD as distance measurements. Visualizations for their latent space by t-SNE (d) KL-divergence, (e) JSD, (f) JTD. Models are trained using vanilla VAE at the VAE layer with 1000 epochs.

$k$. Through the obtained parameter $\alpha_{ik}$ and $S_i$, which is the sum of total evidence and class $K$, the cross-entropy loss changes as follows:

$$\mathcal{L}_i = \sum_{j=1}^{K} y_{ij}(\psi(S_i) - \psi(\alpha_{ij})),  \tag{5}$$

Where $\psi(\cdot)$ is a digamma function, and after this process it has the same effect as specifying the class of uncertainty. Also, we add a regularizing term for misleading evidence. For predicted parameters $\alpha_i$ with true class j, misleading evidence is represented as $\alpha_{ik}$ for all $k \neq j$. With equation 5, we

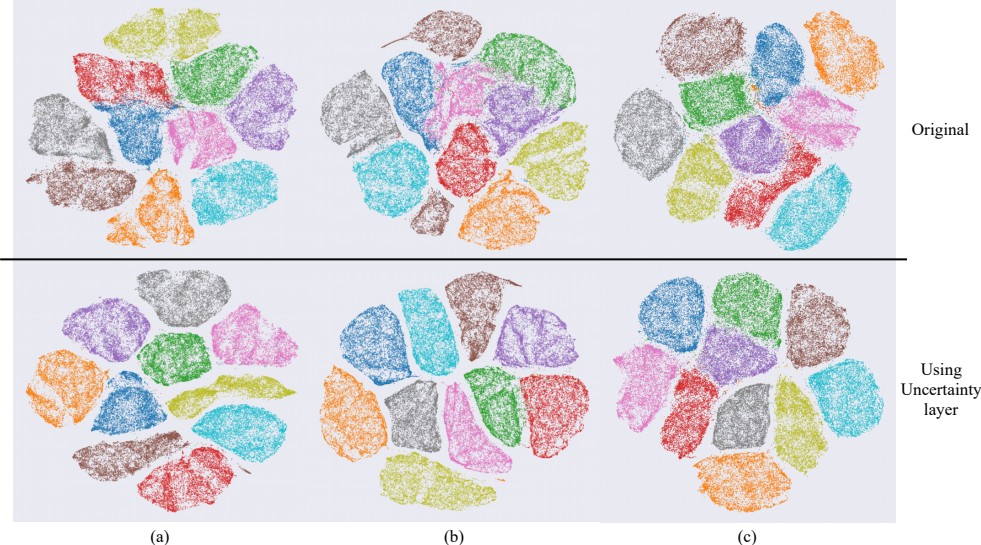

Figure 10: T-SNE visualizations for the embedded FashionMNIST using three types of VAE for the VAE layer. (a) vanilla VAE, (b) $\beta$-VAE, (c) FactorVAE.

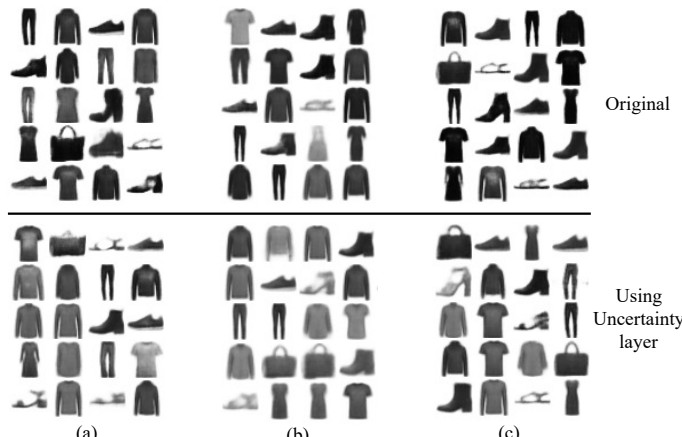

Figure 11: Visualized prototypes from proposed model with three types of VAE for the VAE layer. (a) vanilla VAE, (b) $\beta$-VAE, (c) FactorVAE.

regularize them using KL divergence with the uniform Dirichlet distribution as follows:

$$\mathcal{L}_{\text{uncertain}} = \frac{1}{n} \sum_{i=1}^{n} \mathcal{L}_i + \frac{\lambda_t}{n} \sum_{i=1}^{n} KL\left[D\left(\mathbf{p_i} \mid \overline{\boldsymbol{\alpha}}_i\right) \| D\left(\mathbf{p}_i \mid \langle 1, \ldots, 1 \rangle\right)\right],$$

where $\lambda_t$ is the annealing coefficient that starts from 0 and becomes 1 over training epoch $t = 50$, $D\left(\mathbf{p}_i \mid \langle 1, \ldots, 1 \rangle\right)$ is the uniform Dirichlet distribution, and $\tilde{\boldsymbol{\alpha}}_i = \mathbf{y}_i + (1 - \mathbf{y}_i) \odot \boldsymbol{\alpha}_i$ for removing non-misleading evidence.

### A.6 Additional results for MNIST experiments

As shown in Figure 12(a), prototypes with the shape of 0 are the closest to the input. Distances to numbers such as 2, 3, 8, 9, which have curved parts similar to 0, are closer than the distance to numbers such as 1, 4, 7, which have sharp shapes. Moreover, in the same class, the distance from the input was similar. Further explanations are given by the relationships between input and prototypes and the relationships between each prototype, as shown in Figure 12(b), (c). We can infer

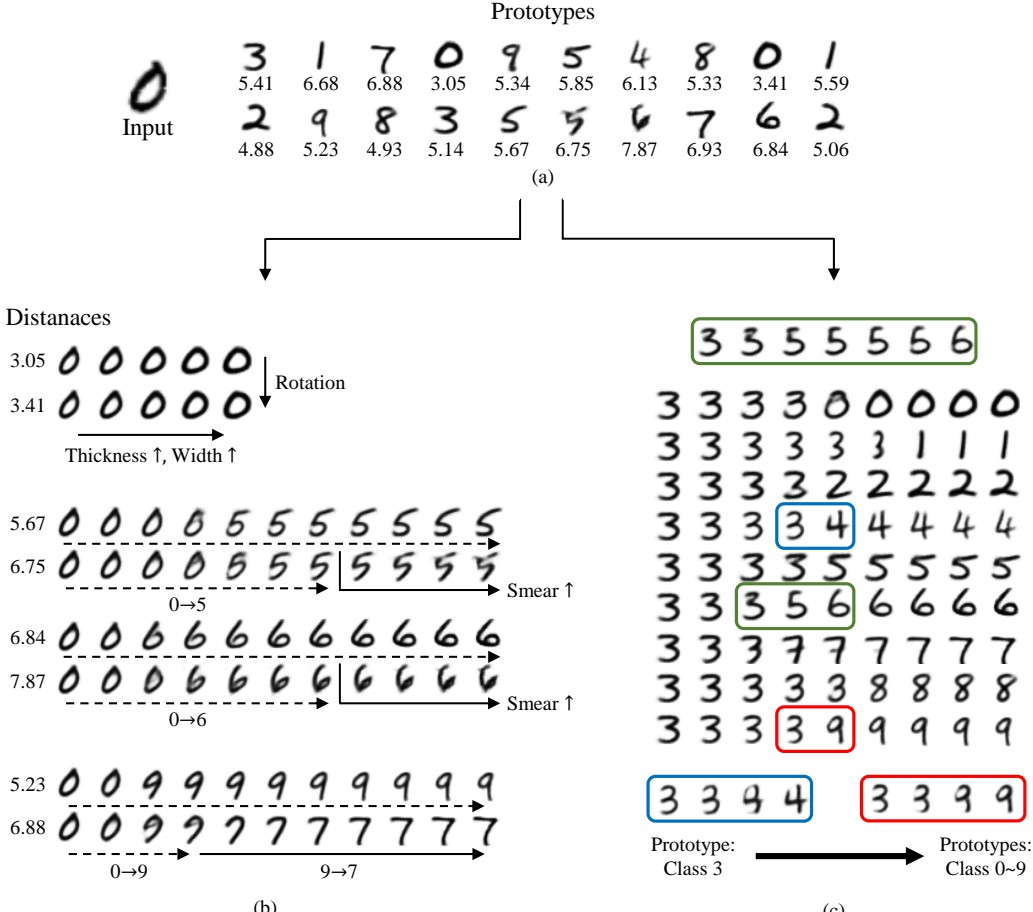

Figure 12: (a) Basic explanation with prototypes and distance between embedded input and prototype distributions. (b) Relationships between input and prototypes. (c) Relationships between prototypes

the factors that caused the model to predict distance farther by the relationships between the input and the prototypes. Also, by examining the relationship between each prototype, we can find out how the model distinguishes the classes. If the explanation is insufficient, more detailed explanations can be obtained by visualizing more interpolation values between prototypes.

## A.7 STABILITY: HOW SHOULD THE EXPLANATION RESPOND TO SIMILAR INPUTS?

Interpretable explanation models have to provide consistent explanations for similar inputs for stability. This is because the reliability of explanations is not guaranteed if different explanations are provided for similar inputs. To demonstrate the stability of our method, we measure the changes in distances provided as an explanation when the input is distorted. For fairness of measurements, the distance change is normalized by distance before the input is distorted. We measure these distance changes for each prototype of all inputs. In Table 5, our method shows much more stability for input distortions. In particular, there is a large difference in terms of average and median in the case of the baseline model, which means explanations that provide significantly different distances exist when the inputs are distorted. Based on our experiment, we claim our method provides a more reliable explanation to users than the baseline model.

Table 5: Distance changes by baseline model, proposed model with VAE, FactorVAE

| Distortion | FactorVAE | | VAE | | Baseline model | |
|---|---|---|---|---|---|---|
| | Median | Mean | Median | Mean | Median | Mean |
| 0.1 | 0.0222 | 0.0294 | 0.0293 | 0.0390 | 0.0701 | 0.1580 |
| 0.2 | 0.0421 | 0.0535 | 0.0595 | 0.0735 | 0.1535 | 0.4666 |
| 0.3 | 0.0502 | 0.0634 | 0.0730 | 0.0923 | 0.1908 | 0.6697 |
| 0.4 | 0.0630 | 0.0784 | 0.0907 | 0.1095 | 0.2575 | 0.1177 |
| 0.5 | 0.0732 | 0.0912 | 0.1044 | 0.1283 | 0.3224 | 1.6922 |

