# OpenReview forum: "Beyond Examples: Constructing Explanation Space for Explaining Prototypes"
_ICLR.cc/2022/Conference — ICLR 2022 Submitted_

### Official Review · Reviewer_RHQ4 · 2021-11-03

**Correctness:** 2
**Technical Novelty And Significance:** 2
**Empirical Novelty And Significance:** 2
**Recommendation:** 3
**Confidence:** 3

**Main Review:**

The continuous nature of the embedding space of a VAE makes it more suitable than an AE. This is an improvement over the previous work of (Oscar Li et al. AAAI 2018).
Training VAE may have several issues: it may require many training samples to be able to generate explanation images of adequate quality. In the paper, generated samples are based on toy datasets MNIST or FMNIST. It is not clear how complicated the image generation process would become if the method be applied to more serious datasets. My concern is that at some point the challenge will become high-quality image generation, rather than a classification task.
Also, the decoder of the VAE may degrade the classification performance (C. Chen Neurips 2019). Too much regularization may degrade the classification performance. Too little regularization (VAE) may result in a poor generation of the explanation images.

Do these "extra" explanations this method is providing offer anything more than a simple prototype? In other words, if the authors had a comparison study in their mechanical turk experiment with PDL (Oscar Li et al. AAAI 2018), would there have been any benefit over using their method? The paper unfortunately hasn't done an experiment with PDL, only comparing to the "no explanation" scenario.

I am not a native English speaker but there are too many grammatical issues in the paper. For example, in pp.2 paragraph 2: “Among our model, (i) relationships …” , Meanwhile, prototypes … are given for explanation.”. “we developed the model that progresses…”. Or in page 9, “it is proper to use …”


**Summary Of The Paper:**

The paper tries to provide an inherently interpretable neural network. Unlike the previous method of (O. Li, AAAI 2018) the paper uses VAE instead of AE. It considers the distance between the distribution of the input and the distribution of prototypes in the latent space of a VAE (the explanation space). The paper also provides a method for determining the number of prototypes using the Bayesian Information Criterion (BIC) for choosing the optimal number of components in the corresponding GMM.


**Summary Of The Review:**

I see values in the paper as using a VAE instead of AE is a natural improvement over previous work. Nevertheless, I have major concerns as described above, in particular, the paper is missing a comparison with previous works especially (Oscar Li et al. AAAI 2018).

---

> ### Author Response · Authors · 2021-11-16
> **Response to Reviewer RHQ4 (2/2)**
>
> \
> **4. The decoder may degrade the classification accuracy. Also, too much regularization may reduce classification accuracy, and too little regularization may not generate images properly.**
>
> The decoder degrades the classification accuracy when the decoder does not generate proper images. As the image is not generated properly, the reconstruction loss does not produce correct values, making it difficult to train the network. This problem can be solved by using a more advanced VAE-based network that can reconstruct well even complex data sets such as NVAE. It is well known that regularization does not necessarily degrade classification accuracy proportionally. The regularization of VAE helps proper compression of the latent space, which induces the representation to place similarly shaped data in a similar position in the latent space. This properly formed latent space actually helps to improve classification performance. In fact, the accuracy was higher when the lambda, the term controlling the classification loss, was set to 50 than when it was set to 100 or 500 in our experiments.
>
> $\lambda = 50: 92.8 \pm 0.1$ (%)
>
> $\lambda = 100: 92.6 \pm 0.2$ (%)
>
> $\lambda = 500: 92.5 \pm 0.1$ (%)
>
> \
> **5. There are too many grammatical issues in the paper.**
>
> Thank you for your pointing out grammatical errors. We clearly recognized that there were many grammatical errors in our paper. We will upload a version with grammatical errors corrected during the discussion period.
>
> \
> Once again, we appreciate you for reviewing our paper, and we hope that misunderstandings and concerns have been resolved. If you have any additional questions or concerns, please let us know. We will do our best to answer them as much as possible.
>
> ---
>
> **References:**
>
> [1] Li, Oscar, et al. "Deep learning for case-based reasoning through prototypes: A neural network that explains its predictions." Proceedings of the AAAI Conference on Artificial Intelligence. Vol. 32. No. 1. 2018.

---

> ### Author Response · Authors · 2021-11-16
> **Response to Reviewer RHQ4 (1/2)**
>
> \
> Thank you for your positive feedback and thoughtful comments.
>
> \
> Your main concern in the review was that there was no comparison with the previous work [1], but there seems to be a misunderstanding here. The main experiments in this paper are made through comparison with PDL [1]. First of all, in the second paragraph of Section 2 Explanation Methods for DNN part, we noted that PDL is used as our baseline model. If you look at 4.1 Reliable Explanation Space in Section 4, which is the most important comparison, you can see the comparison between our method and the latent space of PDL. We compared latent space qualitatively through t-SNE visualization and quantitatively through interactive reconstruction experiments. Through these comparisons, we showed that our method provides more interpretable latent space than PDL, and that distance has "intrinsic meaning" in how it provides prototype and distance as explanations. In the case of our method, we provide a basic explanation in the same way as in previous prototype-based methods and additional explanations, which is the unique and novel explanation of our paper. Section 4.1 shows that our method improves interpretability compared to PDL even in basic explanation (prototype and distance). A comparison of accuracy is also provided in section 5.1.
>
> \
> **1. There is no comparison with the PDL whether the “extra” explanation provided by the paper provides a better explanation than a simple prototype. Mechanical Turk experiment seems to compare only “no explanation” scenario.**
>
> There seems to be a misunderstanding here as well. The method presented in this paper provides not only the prototypes and distances as a basic explanation but also an additional explanation, as shown in Figure 1. Since the previous prototype-based explanation methods provide explanations through prototypes and distances, we have extended the prototype-based explanations into the combination of basic explanations and additional explanations. In this situation, the Mechanical Turk experiment of section 5.2 compared the difference between the case when only a basic explanation (distances and prototypes) was given instead of “no explanation” and the case when an additional relationship explanation was given as described in the third paragraph of section 5.2. These correspond to scenarios with and without relationships. The scenario without relationships is the case where only the prototype and distance are given, such as the mentioned PDL. The reason for this comparison is that, as shown in section 4.1, our method provides more interpretable distances than the PDL, so we conducted a Mechanical Turk experiment in section 5.2 to show the advantage of relationship explanations.
>
> In summary, the comparison with PDL qualitatively and quantitatively showed that our method exhibits more interpretable distances in the explanation method that provides distance and prototype in section 4.1. Therefore, the experiment in 5.2 was conducted to show whether the presence or absence of additional explanations can help people to understand the model's prediction.
>
> It seems that our statement of the description for the Mechanical Turk experiment was not clearly described in the third and fourth paragraphs of 5.2. During the discussion period, we will upload the revised version so that the statement will be more clear.
>
> \
> **2. Training VAE requires many training samples to generate images.**
>
> We agree that many training samples might be required to train the VAE. However, we believe that there's more advantage of learning the latent space to be denser by using VAE. Because distance is the core explanation metric in the prototype-based explanation method, it is important to make the distance similar to the person's perception and show how distance is composed by showing the relationship. However, we understand what you have pointed out and we'll add a discussion of this limitation in the last paragraph of section 4.3 where we wrote about our model's disadvantage.
>
> \
> **3. The model will not be able to generate images properly from images that are more difficult than MNIST and FashionMNIST.**
>
> We agree that our method has these limitations, as noted in the last paragraph of section 4.3. However, this is a problem due to the performance of the VAE model itself. The main novelty of our paper is that it can help the understanding of the deep learning model by visualizing the relationship/similarity and solving the problem of the previous prototype-based DNNs, in which the similar latent representations do not guarantee similarity in shape. We would appreciate it if you reconsider this point. The performance limitations of VAE can be resolved by using large-scale VAEs. If you consider the recent VAE papers, such as the NVAE we cited in the last paragraph of section 4.3, it can be observed that recent VAE models generate the same high-quality image creation as CIFAR-10, Imagenet, and CelebA HQ in 224X224.

---

> ### Author Response · Authors · 2021-11-23
> **About additional experiment**
>
> We would like to inform you about an additional experiment.
>
> Although we performed comparisons with PDL through latent visualization and interactive reconstruction experiments in section 4.1, **Reviewer QGcu**'s suggestion allowed us to proceed with additional quantitative comparisons of stability.
>
> We would really appreciate it if you could refer to the experiment on the stability of explanation in the answer to **Reviewer QGcu**.

---

### Official Review · Reviewer_HXZp · 2021-11-03

**Correctness:** 3
**Technical Novelty And Significance:** 2
**Empirical Novelty And Significance:** 2
**Recommendation:** 5
**Confidence:** 3

**Main Review:**

The key issue that explanation spaces try to address is that having similar latent representations does not guarantee that images share similarities in human-discernible ways, so the method uses VAE to regularize the latent representations. This is an interesting idea.

I have a few questions about the experimental results:
- How was reliability measured (that FactorVAE forms a more reliable explanation space) in Section 5.1?
- It looks like there are two orange clusters in Figure 3b?

**Summary Of The Paper:**

The paper extends prototypes-based explanations by proposing what they call explanation spaces describing the relationships between input and prototypes, the relationship between prototypes, and how prototypes are distributed. They construct explanation spaces using VAEs and suggest a way to find the optimal number of prototypes.


**Summary Of The Review:**

The idea is interesting but not a lot of novelty, as some existing papers (e.g. Li's Deep Learning for Case-Based Reasoning through Prototypes paper) already provide notions of distances between inputs and prototypes, prototypes and test data points, etc.

---

> ### Author Response · Authors · 2021-11-16
> **Response to Reviewer HXZp**
>
> \
> Thank you very much for reviewing our paper thoughtfully.
>
> The following are our comments on your thoughtful comment.
>
> \
> **Summary Of The Review: The idea is interesting but not a lot of novelty, as some existing papers (e.g. Li's Deep Learning for Case-Based Reasoning through Prototypes paper) already provide notions of distances between inputs and prototypes, prototypes and test data points, etc.**
>
> Prototype-based explanation AI models that provide distances and prototypes are being studied in various ways, as described in Section 2 Explanation Methods for DNN. We solved the problem that distances do not guarantee a similar appearance, which is a common problem of the previous prototype-based explanation models. Furthermore, we made it possible to provide an additional explanation of the relationship to resolve users’ doubts about the explanation of the distance.
>
> On the part of understanding the model, we propose an additional method to describe similarity/relationship, going beyond the previous methods that provide prototypes and distances as explanations. It was shown through the Mechanical Turk experiments in section 5.2 that this method helps users’ understanding. Rather than only providing distances and prototypes, it is possible to better understand a model by showing which features are changing through relationships, so that users are able to understand how distances are composed. Furthermore, when distances and prototypes are provided as explanations, the distance has an “intrinsic meaning” as a numerical value that can guarantee similarity in our method, giving more accurate explanations for the user to understand the model. If the distance does not guarantee the similarity perceived by humans, it cannot serve as a sufficient explanation for understanding the model. It is emphasized that this is a key contribution. This part is also mentioned by **Reviewer bDHd**,
> > “Their model uses VAE to ensure similar latent representations share similarities in appearance, a problem that many existing prototype-based CNN models suffer.”
>
> Therefore, extending the example-based prototype into the explanation space-based prototype provides a better explanation with a more interpretable representation model for humans.
>
> Regarding technical parts, we made the latent space more interpretable through the VAE and VAE training techniques. Since VAE encodes an image as a Gaussian distribution, we changed a vector of prototype to a distribution of prototype. Instead of the L2 distance, we computed an appropriate 2-Wasserstein distance to compare the distances between distributions and added an uncertainty layer at the end of the network. The major technical contributions of the paper therefore include: 1) to extend input encoding vectors and prototype vectors of [1] into input encoding distributions and prototype distributions, and 2) to use VAE to regulate the latent space so that distances in the latent space can be explained visually.
>
> In addition, we wrote in reply to **Reviewer QGcu**'s question 1 about the problem that our paper was trying to solve. We would appreciate it if you read it once. Please understand our novelty with the above contents.
>
> \
> **1. How was reliability measured in Section 5.1?**
>
> We measured the interpretability of the latent space in an interactive reconstruction experiment [2] with the same settings as presented in 5.1 and appendix A.2. It is an experiment to make an image similar to a randomly sampled image by changing each dimension value at a randomly given position in the latent space. The generated image is created through the decoder, and as the user explores the latent space and generates a similar image, you can see how easy to understand the latent space for the user. By looking at the success rate of the user creating a similar image, you can see how intuitively the configuration of the latent space is understood by the user, which means how intuitively the distance in the latent space is understood by the user. As mentioned in [2], a higher success rate means higher interpretability that comes to higher reliability.
>
> \
> **2. It looks like there are two orange clusters in Figure 3b?**
>
> It is true. Even in Figure 3(b), although not as segregated as in Figure 3(a), a separate cluster exists. Therefore, we applied FactorVAE to regulate the latent space more. The result is shown in Figure 3(c), and this difference in latent space is similar to the difference in the success rate in Table 1.
>
> ---
>
> **References:**
>
> [1] Li, Oscar, et al. "Deep learning for case-based reasoning through prototypes: A neural network that explains its predictions." Proceedings of the AAAI Conference on Artificial Intelligence. Vol. 32. No. 1. 2018.
>
> [2] Ross, Andrew, et al. "Evaluating the Interpretability of Generative Models by Interactive Reconstruction." Proceedings of the 2021 CHI Conference on Human Factors in Computing Systems. 2021.

---

> ### Author Response · Authors · 2021-11-29
> **Additional Response**
>
> We would like to inform you about an additional experiment and our introduction modification.
>
> \
> We revised the whole introduction section to state the novelty of our paper. The update shows that our paper can provide human-interpretable and reliable explanations than existing prototype-based models. Regarding our novelty, the **Reviewer QGcu** said:
>
> >Such construction ensures that the network is actually comparing the distances between distributions to reason about the prediction, where in the previous instance-based explanations, the distance between the input and the prototype is enforced to do so.
>
> \
> Since prototype-based DNNs are mostly based on the method of Li's Deep Learning for Case-Based Reasoning through Prototypes paper, we performed comparisons through latent visualization and interactive reconstruction experiments in section 4.1 in our first draft. In addition to these comparisons, **Reviewer QGcu**'s suggestion allowed us to proceed with additional quantitative comparisons of stability. The manuscript to be added is in the "Additional Response" to **Reviewer QGcu**.
>
> \
> We would really appreciate it if you could take these modifications into consideration.

---

### Official Review · Reviewer_bDHd · 2021-11-04

**Correctness:** 4
**Technical Novelty And Significance:** 3
**Empirical Novelty And Significance:** 3
**Recommendation:** 8
**Confidence:** 3

**Details Of Ethics Concerns:**

I don't see any ethics concerns

**Main Review:**

This paper improved existing prototype-based DNN by enhancing interpretability and while also improving the performance. I enjoyed reading the paper and I think the proposed solutions will benefit other prototype-based DNN models for image classification. I have a couple of questions that I wish the authors could answer.

1) in the original ProtoPnet paper, the authors Li et al designed separation cost and clustering cost in their objective, with the goal to push images closer to prototypes of the same label, and away from ``incorrect'' prototypes. I don't see such designs in this paper. Is there a reason that you don't choose to include these terms?

2) In some prototype-based models, they also include a \emph{diversity} term, which encourages protoypes to be different, to avoid redundancy. There are no such designs in the proposed model. Do prototypes have redundancy issues? do some of them look very similar to each other?

3) I'm wondering if you can discuss the connection between your explanations for basic explanations and counterfactual explanations, because it seems by changing certain features, following your basic explanations and explanations for basic explanations, you can guide the model to get a different prediction. You did mention that "Visualized images of our explanations are fundamentally generated through VAE, making it challenging to generate human-recognizable images for complex datasets that are difficult for VAE to generate the observable image." Would that be a reason that your model cannot be used to generate counterfactual explanations?

**Summary Of The Paper:**

This paper extends existing series of prototype-based DNN for image classification. Their model uses VAE to ensure similar latent representations share similarities in appearance, a problem that many existing prototype-based CNN models suffer; They also include an uncertainty layer which improves the predictive performance. The authors also conducted human evaluations via Amazon Mechanical Turks to validate the interpretability of their model

**Summary Of The Review:**

Overall I enjoyed reading the paper. I only have a few questions, which basically asked the authors to clarify their design choices, especially those that are different from existing works.

---

> ### Author Response · Authors · 2021-11-16
> **Response to Reviewer bDHd**
>
> \
> First of all, we appreciate you very much for reading and reviewing our paper.
>
> Thank you for recognizing the problems in the previous prototype-based papers and accurately summarizing our contribution.
>
> \
> **1. Why not use separation loss and clustering loss?**
>
> The reason for clustering loss and separation loss are used is that if the input data is trained only through the classification loss, data belonging to the same class in the latent space are not clustered well. In this case, there are cases where the prototype does not actually represent the data. On the other hand, our method uses VAE and does not use separation loss and clustering loss because data with similar shapes are naturally aggregated in latent space. If you look at Figure 3, you can see that the clustering of the latent space is actually well done.
>
> \
> **2. Why not use diversity loss? Are there duplication issues where parts of your prototype are very similar to each other?**
>
> We know there are papers like [1] that use diversity loss for separation between prototypes. However, when the diversity term between prototypes is used, the prototype formed in the latent space to be suitable for classification is dropped to an arbitrary intensity through a hyper-parameter. We did not use the diversity term because it could place a prototype in an undesirable location when the latent space was regulated as a Gaussian prior.
>
> If you increase the number of prototype distributions excessively, there are cases where parts of the prototypes look very similar to each other, but if you use the right number of prototypes, there is no prototype duplication problem. In this process, to find a suitable number of prototypes, we found a method like section 4.2.
>
> \
> We know the losses you mentioned are widely used appropriate losses that can be applied to our method, but since VAE forms the latent space to work well even in the absence of the losses with reasons mentioned earlier, we did not use the loss terms that need to be adjusted through additional hyper-parameters.
>
> \
> **3. I'm wondering if you can discuss the connection between your explanations for basic explanations and counterfactual explanations.**
>
> Thank you so much for giving us an interesting point of view. Since our proposed method creates a dense latent space, it is thought that it will be able to generate counterfactual explanations sufficiently. Because our model uses VAE, we can form an appropriate image in any part in the latent space compressed with Gaussian prior distribution. Since all the processes of approaching prototypes from input data can be reused as input data, it is possible to make the model predict differently while checking the relationship with prototypes of other classes. We can generate counterfactual explanations from our model's explanations for basic explanations.
>
> The disadvantages you mentioned were issues with the performance of the underlying VAE itself. Therefore, sufficient creation is possible in the MNIST and FashionMNIST datasets. Also, using a large-scale VAE model such as [2] mentioned in the last paragraph of section 4.3 can generate high-quality images even in complex data sets, so it is expected that counterfactual explanations can be generated even in complex data sets.
>
> We would like to conduct new research in this direction. Thank you very much for suggesting a topic for future research.
>
> ---
> **References:**
>
> [1] Gee, Alan H., et al. "Explaining deep classification of time-series data with learned prototypes." CEUR workshop proceedings. Vol. 2429. NIH Public Access, 2019.
>
> [2] Vahdat, A., & Kautz, J. (2020). NVAE: A deep hierarchical variational autoencoder. Advances in Neural Information Processing Systems, 2020-December(NeurIPS), 1–13.

---

### Official Review · Reviewer_QGcu · 2021-11-06

**Correctness:** 2
**Technical Novelty And Significance:** 3
**Empirical Novelty And Significance:** 2
**Recommendation:** 5
**Confidence:** 2

**Main Review:**

Overall the paper introduces an interesting concept, *explanation space*, and an interesting way of constructing networks that incorporate some constraints, i.e. the distance layer, into the model. As a result, part of the internal components are not black boxes to humans any more. It also comes with an easier way to visualize the internal activations of the model by the decoder part of a VAE.

However, even though I appreciate the potential merits in the paper’s idea of building better networks in terms of interpretability, I have several concerns about the motivation of the technique and inadequate evaluations that may not be sufficient to convince the reader. Please find my detailed review as follows.

### Novelty

The idea of combining distance layer with VAE is novel and the concept of *explanation space* is also first proposed in this paper. These two techniques make this paper stand out from other VAE-based explanations. However, these techniques are not well-motivated to me and the particular application of the proposed visualizations, explanation space, is also missing. To clarify, I have two following questions regarding the motivation of the paper:

1. What is the concrete problem this paper tries to solve? As the authors argue that “However, examples are powerful but not enough” in the introduction of the paper, the sentence seems not to be completed because the most important object is missing -- the example-based explanations are not enough for what (questions)? I am interested in what kind of questions that the authors have in mind that a human user may have but is not able to fully answer by the current explanation approaches. For example, with a classifier that differs dogs from cats, what does the concrete question look like, which requires a set of explanations, i.e. explanation space, instead of examples generated by prior work. Without a clarification like that, I am not sure how the evaluations provided in the result of the paper justify the significance and validate the efficiency of the proposed approach.

2. The current motivation for using the uncertainty layer seems to be quite weak. The authors do not justify what motivates the use of an uncertainty layer on the top of the network. The only relevant information I am able to find is that by using the uncertainty layer the accuracy of the model is improved by 1%. This is not convincing enough to serve as a strong motivation because :1) the improvement of performance is strongly related to the data distribution, which does not serve as a motivation if using other datasets. For example, 1% increase can be very significant for larget datasets, i.e. ImageNet, but pretty trivial for the datasets used in this paper, FashionMNIST and MNIST. 2) the improvement of the accuracy should instead be the result of employing the uncertainty layer instead of the motivation to do so. The way the current paper justifies the use of this layer does not convince me of the necessity of it. And i am curious to see how the changes in explanations may look like with the 1% decrease of the network’s performance.

A minor point: understanding the model’s internal behavior by the distances of representations learned by VAE or any encoder-decoder networks are not first proposed in this paper . Some related work [1] seems worth mentioning and discussing.


### Technical Quality

I have three concerns regarding the analysis and the experiments in this paper.

1. The empirical findings seem to be the main contributions of this paper, however, some descriptions seem to convey the message that the proposed explanation enjoys some nice properties but they are either not clearly defined or proved. As the authors emphasize in the abstract that “... but we propose an inherently interpretable model for more faithful explanations.”, I am not able to locate clear definitions in the paper about “inherently interpretable” and a measurable definition of “faithfulness” to support the argument that the proposed method is **more** faithful. Further, what are the baselines for the authors to derive the conclusion that the proposed method is **more faithful** (I will come back to the baseline issue in the next bullet point).

2. The empirical evaluations are not strong enough to support the claims that the authors make about the contributions. Firstly, most of the experiments are conducted on MNIST and FashionMNIST but these two data distributions are sometimes too simple to generalize to higher dimensions, i.e. ImageNet. It is not new to the community that a lot of algorithms will work perfectly on MNIST and FashionMNIST but fail dramatically on more complicated colorful images as in practice you may not actually need a deep net to achieve a great performance on MNIST. For example, the K-nearest neighbors classifier is both explainbale and accurate on MNIST. I would encourage the authors to expand the empirical evaluations to datasets with higher dimensions in the input features before rushing to make any conclusions.

3. Baselines are missing in the evaluation section. The major part of the evaluation section seems to focus on explore that the proposed method can bring to the user. However, before rushing into the exploration part of the method, there is a missing part of comparing the proposed method with the prior work. Even though the authors emphasize that the proposed method provides a set of examples instead of just one, which does not exist in the previous work. However, it does not seem to be unfair to compare one or several examples samples from the explanation space with some prior work that provides instance-based explanations [2, 3, 4, 5, 6] unless I misunderstand the paper and please help me understand why such comparisons are not useful to show the proposed explanation space provides better explanations. These baselines may not provide an apple-to-apple comparison but are worth discussing and performing some comparisons in the ballpark.


### Clarity
The writing of the introduction section is a bit confusing to me because I miss the part about what the problem the paper aims to answer and how the results in the evaluation sections help to show that the proposed method solves the problem. Figure 1 and 2 are clear to help readers to understand the proposed architecture and I appreciate that.

### Significance

With the aforementioned review, the significance of this paper can vary. One the one side, if the authors can help me understand the what is the subject of the explanation, that is the question requiring an answer like the one proposed in the paper and can not be answered by prior work, and why the current evaluation is sufficient to support the claim, I would recommend this paper because the contributions are significant to the explanation community. On the other hand, the current manuscript does not seem to be able to convince me that the contributions are sound and significant. Therefore, I am on the negative side but will decrease my confidence in the score because I am willing to increase it once my concerns are resolved.


[1] Yang, Ceyuan et al. “Semantic Hierarchy Emerges in Deep Generative Representations for Scene Synthesis.” Int. J. Comput. Vis. 129 (2021): 1451-1466.

[2] Pang Wei Koh and Percy Liang. 2017. Understanding black-box predictions via influence functions. In Proceedings of the 34th International Conference on Machine Learning - Volume 70(ICML'17). JMLR.org, 1885–1894.

[3] Kim, Been et al. “Examples are not enough, learn to criticize! Criticism for Interpretability.” NIPS(2016).

[4] Yeh, Chih-Kuan et al. “Representer Point Selection for Explaining Deep Neural Networks.” NeurIPS(2018).

[5] Pruthi, Garima et al. “Estimating Training Data Influence by Tracking Gradient Descent.” ArXivabs/2002.08484 (2020): n. Pag.

[6] Goyal, Yash et al. “Counterfactual Visual Explanations.” ArXiv abs/1904.07451 (2019): n. pag.


**Summary Of The Paper:**

The main contributions of this papa are 1) a particular architecture that combines Variational Auto-Encoder (VAE) that makes the predictions based on the distances between instances and proto-types; and 2) a series of visualizations in the latent space that explains the model’s predictions, i.e. explanation space. In order to verify their assumptions that their networks provide faithful and rich explanations, the authors conduct user studies on MNIST and FashionMNIST datasets.

**Summary Of The Review:**

Overall I recommend for a rejection because the current version of the paper is not well-motivated and the evaluations are not sufficient to support the claims and contributions made by this paper. I am open to any discussions and will increase my score if my concerns are resolved.

---

> ### Author Response · Authors · 2021-11-16
> **Response to Reviewer QGcu (3/3)**
>
> \
> **4. The empirical evaluations are not strong enough to support the claims that the authors make about the contributions.**
>
> Whether it works well with high-dimensional data is a question of the performance of the VAE model itself, and we think that it can be solved by using a large-scale VAE model. The advantage of our method is that it is a result obtained through the formation of a dense latent space by VAE, so it is a result that can be obtained in common in models with VAE structure. In the case of recent papers such as NVAE [6] cited in the last paragraph of section 4.3, which mentioned the disadvantages of our paper, it can be confirmed that it works well even with high-dimensional datasets. In the NVAE paper, you can see that it has the properties of VAE and can accurately generate high-quality images like Imagenet, CIFAR-10, CelebA HQ of 224X224.
>
> The focus of our paper is to visualize the relationship between prototypes and to solve the problem that similar latent representations do not share similarities in appearance common in previous prototype-based DNNs for image classification. These contributions are ultimately to help users understand the judgment of the model. Therefore, we conducted the experiment focusing on evaluating how helpful our method was for the explanation of the model, and we think it can be sufficiently verified by MNIST and FashionMNIST. We would like to mention that resolving image generation in large data sets is out of scope for our work, and it could be suitable for our future work.
>
> \
> **5. Baselines are missing in the evaluation section.**
>
> In our paper, the comparison with prior work is written in section 4.1. As mentioned in the Introduction, we wanted to develop an inherently interpretable model. Among these inherently interpretable models, we wanted to create a model that provides explanations through examples that are known as powerful explanations. That's why we created a prototype-based explanation model, as shown in Section 2 Explanation Methods for DNN.
>
> In this process, since prototype-based DNNs are mostly made based on the method of [7], there is a problem that similar latent representations do not share a similar appearance. We developed a model to solve these common problems and expand prototype-based explanations. Therefore, as shown in the second paragraph of Explanation Methods for DNN in Section 2, we compared with [7], the paper used as the basis for prototype-based DNN, as a baseline.
>
> Therefore, comparison with the method used as the baseline was carried out in section 4.1, and comparison with methods corresponding to the post-hoc explanation was not carried out. After comparison with the prior work, it was shown that the additional explanations obtained with our method aided the understanding of the judgment of the model in section 5.
>
> \
> **Minor point: About related work [8]**
>
> It is interesting to learn by separating factors that can control image generation by using the latent space of GAN. It seems to be related to [9], which we cited, so we will add it to related work.
>
> \
> **Clarity**
>
> We would like to add a phrase that expresses the strength of our method well in the introduction part. Also, it doesn't seem to be well expressed in our paper that the additional explanation helps users to understand the prediction of the model. Therefore, as we have mentioned in question 1, we will include descriptions through examples in the introduction. The paragraphs related to the Mechanical Turk experiments in Section 5.2 will be revised to be neater. These modifications will be completed within the discussion period.
>
> ---
> **References:**
>
> [1] Javier Antoran, U. B. (2021). GETTING A CLUE: A METHOD FOR EXPLAINING UNCERTAINTY ESTIMATES. ICLR 2021.
>
> [2] Rudin, C. (2019). Stop explaining black-box machine learning models for high stakes decisions and use interpretable models instead.
> Nature Machine Intelligence.
>
> [3] Arik, S. O., & Pfister, T. (2020). Protoattend: Attention-based prototypical learning. Journal of Machine Learning Research.
>
> [4] Ghorbani, A., Zou, J., Wexler, J., & Kim, B. (2019). Towards Automatic Concept-based Explanations. NeurIPS.
>
> [5] Moraffah, Raha, et al. "Causal interpretability for machine learning-problems, methods and evaluation." ACM SIGKDD Explorations Newsletter 22.1 (2020).
>
> [6] Vahdat, A., & Kautz, J. (2020). NVAE: A deep hierarchical variational autoencoder. Advances in Neural Information Processing Systems, 2020(NeurIPS).
>
> [7] Li, Oscar, et al. "Deep learning for case-based reasoning through prototypes: A neural network that explains its predictions." Proceedings of the AAAI Conference on Artificial Intelligence. Vol. 32. No. 1. 2018.
>
> [8] Yang, Ceyuan et al. “Semantic Hierarchy Emerges in Deep Generative Representations for Scene Synthesis.” Int. J. Comput. Vis. 129 (2021): 1451-1466.
>
> [9] Lang, Oran, et al. "Explaining in Style: Training a GAN to explain a classifier in StyleSpace." arXiv preprint arXiv:2104.13369 (2021).

---

> > ### Comment · Reviewer_QGcu · 2021-11-21
> > **Some Follow-ups**
> >
> > I appreciate the authors' efforts on updating the paper and their responses. By reading through the authors' response, I think the goal of the paper is to introduce a new kind of neural network that is aware of the distances between the input distribution and the prototype distributions. Such construction ensures that the network is actually comparing the distances between distributions to reason about the prediction, where in the previous instance-based explanations, the distance between the input and the prototype is enforced to do so. The motivation of the paper is now clearer to me and I appreciate that.
> >
> > My concerns about the evaluations remain at this point. I will clarify.
> >
> >
> > **Scalability of the approach**
> >
> > All these empirical findings are from MNIST and FashionMNIST and by showing VAE is able to do image reconstruction (as quoted from the authors' response) it is not convincing to me that this is enough to show the proposed training will scale up to ImageNet easily in an obvious way.
> >
> > >  In the NVAE paper, you can see that it has the properties of VAE and can accurately generate high-quality images like Imagenet, CIFAR-10, CelebA HQ of 224X224.
> >
> > My argument is based on that even for training MNIST and FashionMNIST, the authors have demonstrated that the need of an uncertainty layer for separating latent representations and tuning the number of prototypes. When the complexity of the data distributions increases, there are a lot of practical challenges people need to solve, which prevents me from rushing to the conclusion that this approach will certainly generalize to large and RGB images.
> >
> > > We would like to mention that resolving image generation in large data sets is out of scope for our work, and it could be suitable for our future work.
> >
> > I am also a bit unclear why generalizing to larger images and complex datasets is out of the scope for validating the scalability of the proposed network structure when there are no particular blockers from training such networks. Moreover, on the current datasets, the differences between KLD, JSD, JTD and WSD are marginal and trivial. I would blame this on the datasets.
> >
> >
> > **Evaluations**
> >
> > The comparison with Li et al. (2018) in Fig. 3 focuses only on the visualization of the latent space, which does not appear to be an evaluation for the quality of the explanation instead of an evaluation for the training. I find the authors include the results from user study, which is useful and valuable. However, I do believe some quantitative evaluations for the explanations themselves are necessary. For example, consider the following questions:
> >
> >  1) What does a good explanation mean for a prototype-based approach?
> >
> > 2) How does/should the explanation respond to similar inputs?
> >
> > 3) Are prototypes found by the explanations truly the images that are most likely to be classified as the target class?
> >
> > I am pretty sure that by answering these questions we should be able to develop at least one quantitative metric to evaluate **the quality of the explanation**, which is more reliable than just visualizing the latent space distributions. The questions I mentioned above are inspired by a similar work [1] in inherently-interpretable model.
> >
> >
> > [1] Alvarez-Melis, David and T. Jaakkola. “Towards Robust Interpretability with Self-Explaining Neural Networks.” NeurIPS (2018).

---

> > > ### Author Response · Authors · 2021-11-23
> > > **Response to Follow-ups (2/2)**
> > >
> > > \
> > > **Evaluations**
> > >
> > > \
> > > Thank you for providing questions for the quantitative evaluation of the explanation itself. We believe that the questions provided can be a good criterion for evaluating what a good explanation is.
> > >
> > > \
> > > **1. What does a good explanation mean for a prototype-based approach?**
> > >
> > > In the prototype-based explanation method, an explanation is given in terms of distance and prototype. A good explanation consists of human-interpretable or human-understandable distance and a prototype that can represent a data set. Since we deal with distances in a latent space, the interpretability of a latent space is the most important factor influencing the explanation. This is because the interpretability of a latent space leads to the interpretability of the distance. Therefore, we show the interpretability of the latent space with Figure.3 and Table.1.
> > >
> > > \
> > > **2. How does/should the explanation respond to similar inputs?**
> > >
> > > As mentioned in [1], which you referenced, similar explanations should be provided for similar inputs. This leads to the stability of the explanation noted in [1]. We conduct additional experiments to demonstrate the stability of our explanation.  In a prototype-based method, the explanation changes through a change in distance as the input changes. After distorting the input, we measure the changes in distances.
> > >
> > > For fairness of measurements, the (normalized) distance change is calculated by |"distance from the prototype obtained from the input" – "distance from the prototype obtained from the distorted input"| / "distance from the prototype obtained from the input". We measure these distance changes for each prototype of all inputs. We also used torchvision's RandomPerspective function to distort the input while maintaining a similar shape. The distance change values obtained in this way are as follows.
> > >
> > > | Distortion | FactorVAE - median | FactorVAE - avg | VAE - median | VAE - avg | Baseline - median | Baseline - avg |
> > > |:----------:|:------------------:|:---------------:|:------------:|:---------:|:-----------------:|:--------------:|
> > > |     0.1    |       0.0222       |      0.0294     |    0.0293    |   0.0390  |       0.0701      |     0.1580     |
> > > |     0.2    |       0.0421       |      0.0535     |    0.0595    |   0.0735  |       0.1535      |     0.4666     |
> > > |     0.3    |       0.0502       |      0.0634     |    0.0730    |   0.0923  |       0.1908      |     0.6697     |
> > > |     0.4    |       0.0630       |      0.0784     |    0.0907    |   0.1095  |       0.2575      |     1.1177     |
> > > |     0.5    |       0.0732       |      0.0921     |    0.1044    |   0.1283  |       0.3224      |     1.6922     |
> > >
> > > From the above results, it is observed that our model shows much more stable results for input distortions. In particular, in the case of the baseline model, there is a large difference in terms of average and median, which means that there are explanations that provide significantly different distances when the inputs are distorted. This may be because, as we claimed, similar latent representations did not guarantee the similarity of shapes in existing models. Based on our experimental results, it is claimed that the explanation provided by our model is more “stable”. Our explanation shows better stability by providing similar distances for similar inputs, and it will be more reliable.
> > >
> > > The above table will be produced as a plot with the X-axis as distortion, and the Y-axis as mean and median. We will add the above description with a plot to section 4.1.
> > >
> > > We greatly appreciate this particular reviewer’s comment because it was helpful for us to create a new quantitative metric, and to evaluate the quality of explanations.
> > >
> > > ---
> > >
> > > **References:**
> > >
> > > [1] Alvarez-Melis, David and T. Jaakkola. “Towards Robust Interpretability with Self-Explaining Neural Networks.” NeurIPS (2018).

---

> > > > ### Comment · Reviewer_QGcu · 2021-11-29
> > > > **Thanks for the additional experiments**
> > > >
> > > > I apologize for my late reply. I appreciate that the authors have added the additional experiments about the stability of the approach, which looks interesting. One more suggestion for presenting the the results is that the it would be better to add the output scores, or some other quantities that can reflect the model's output behavior, together with the distortions made to the input because essentially we want a faithful explanation -- namely, the explanation should be reasonably sensitive to the model's actual behavior. If the distortion has changed the input a lot and the model is not able to make the similar decision after that distortion, I believe the explanation should follow that change and therefore it is not reasonable to ask that the explanations are still very similar to undistorted inputs.
> > > >
> > > > I appreciate the discussion with the authors. As we approach to the end of the discussion period, I will increase my score from 3 to 5 to reflect the author's effort on the paper and our discussions. This paper is very interesting and could deliver insights and future directions to the XAI community. I would give even higher scores if any RGB datasets (e.g. Tiny ImageNet, ImageNet, Stanford Dogs, Flowers; CIFAR sometimes might actually be a bad dataset for visualizations because it is too small and we barely see the objects) are included in the paper.

---

> > > > > ### Author Response · Authors · 2021-11-29
> > > > > **Thanks for the additional response**
> > > > >
> > > > > Thank you very much for your additional response and raising the score.
> > > > >
> > > > > We agree that the explanation should be reasonably sensitive to the model’s actual behavior. However, the most important aspect of our experiment is the difference between the mean and the median rather than each value. The significant difference between the mean and the median values indicates that there exist values with unusual behavior. This means when the input is distorted, the change in the explanation becomes inconsistent, which leads to an unstable explanation. This does not mean that the explanation with distorted inputs should be similar to the one with undistorted inputs.
> > > > >
> > > > > As we understand and agree your comments, we will revise the manuscript in the “Additional Response for Reviewer QGcu” focusing on the difference between the mean and the median values, not on each value, and provide a more accurate definition of stability through the above. To provide accurate information, we will experiment with appropriate distortion values through accuracy decrease (also accuracy decrease value will be added).
> > > > >
> > > > > We appreciate you again for your helpful feedback and comment that our paper can provide insight and future direction to the XAI community.

---

> > > > > ### Author Response · Authors · 2021-11-30
> > > > > **Additional experiments with accuracy drop & mean/median ratio**
> > > > >
> > > > > As the reviewer mentioned, the small value of the normalized distance change may not mean that the explanation is stable, but the small variance of explanation changes can indicate stability. The large variance of explanation changes means that output distance changes resulting from the same intensity of input distortion differ significantly from each other, and therefore that the stability of explanation is low. The variance of explanation changes can be confirmed through the mean-median ratio. The results of measuring the mean-median ratio and accuracy drop with input distortion are as follows.
> > > > >
> > > > > | Distortion | FactorVAE -  mean/median ratio | FactorVAE -  acc drop (%) | VAE -  mean/median ratio | VAE -  acc drop (%) | Baseline -  mean/median ratio | Baseline -  acc drop (%) |
> > > > > |:----------:|:------------------------------:|:-------------------------:|:------------------------:|:-------------------:|:-----------------------------:|:------------------------:|
> > > > > |    0.05    |              1.21              |            0.04           |           1.29           |         0.03        |              1.75             |           0.02           |
> > > > > |     0.1    |              1.33              |            3.76           |           1.37           |         3.70        |              2.25             |           4.95           |
> > > > > |    0.15    |              1.29              |           10.33           |           1.35           |        10.32        |              2.60             |           12.38          |
> > > > > |     0.2    |              1.27              |           18.13           |           1.34           |        17.30        |              3.04             |           20.38          |
> > > > > |    0.25    |              1.26              |           26.89           |           1.33           |        26.62        |              3.50             |           29.37          |
> > > > >
> > > > > Experimental results show that the variance of explanation changes in our model is small, which means that explanation changes in proportion to input distortion. Based on additional experiments, we believe that our method provides a more stable explanation than the baseline model because the explanation rarely changes drastically with input distortion. In addition, it can be seen that our model shows more stable mean-median ratios than the baseline model, from the case where the accuracy drop is small to the case where the accuracy drop is large. Additionally, our model shows a lower decrease in accuracy for distortion.

---

> > > ### Author Response · Authors · 2021-11-23
> > > **Response to Follow-ups (1/2)**
> > >
> > > Thank you very much for the response, and thank you for mentioning our motivations.
> > >
> > > Our answers about concerns are as follows.
> > >
> > > \
> > > **Scalability of the approach**
> > >
> > > \
> > > **1. It is not convincing to me that this is enough to show the proposed training will scale up to ImageNet easily in an obvious way.**
> > >
> > > Our method proceeds classification and provides explanations through distances in the latent space encoded by VAE's encoders. Good image generation in VAE-based networks means that the encoder has a well-formed dense latent space (this is because we set the latent representation in the latent space and generate the image through the decoder with this value). In our method, it is equivalent to setting anchors called prototypes in this latent space. Through these anchors, the latent space is transformed into a more disentangled form. NVAE performs image generation well even on complex data sets and forms dense latent spaces well. Since the proposed method sets anchors called prototypes in a latent space and performs classification and explanation via the distance from these anchors, we thought that the VAE structure network forming a dense latent space would scale up reasonably well in complex data sets.
> > >
> > > \
> > > **2. My argument is based on that even for training MNIST and FashionMNIST, the authors have demonstrated that the need of an uncertainty layer for separating latent representations and tuning the number of prototypes.**
> > >
> > > Our uncertainty layer is not implemented by changing the structure of the model, but by transforming the calculation of the last softmax layer. Therefore, the uncertainty is calculated with the distance values of a latent space, which is processed by a fully connected layer. Even in complex datasets, this method can be applied well if the latent space is well-formed. Please note that an uncertainty layer may not be needed for simple datasets, but may be required in situations where a VAE structure constructs a latent space with sampling regardless of the complexity of data.
> > >
> > > We also suggest how to find the optimal number of prototypes, rather than fine-tuning the number of prototypes needed to work. As shown in section 4.2 selecting the number of prototype distribution, in the cases of the existing methods, a heuristic value determined by a human as the number of prototypes. Instead of a heuristic method, we proposed a method of finding an optimal value for users.
> > >
> > > \
> > > **3. I am also a bit unclear why generalizing to larger images and complex datasets is out of the scope for validating the scalability of the proposed network structure when there are no particular blockers from training such networks.**
> > >
> > > We also agree that the scalability of the proposed network structure is necessary. However, please note that the scalability issue can be dealt with as in Answer #1. Since our paper showed the advantages of using the latent space of a VAE structure in prototype-based explanation methods, we also believe that ImageNet, a larger image dataset such as the one you suggested, is a natural next step.
> > >
> > > Since the scalability issue is important, we will mention the same comment as Answer #1 in the manuscript. We will add this to the last paragraph of section 4.3, which deals with the limitations of our paper.

---

> > > ### Author Response · Authors · 2021-11-28
> > > **Additional Response for Reviewer QGcu**
> > >
> > > Since the discussion period for updating the ICLR draft has passed, we would like to share with you how the manuscript would be updated according to new experiments.
> > >
> > > * We will change the title of section 4.1 to “Reliable Explanation” because, by adding experiment, section 4.1 can more clearly show that a reliable explanation can be obtained by our method.
> > >
> > > * We will group existing paragraphs in section 4.1 into "Interpretable Explanation Space" through \paragraph{} as in section 2 “Explanation Methods for DNN”. In addition, the 2nd and 3rd paragraphs, which are about the interactive reconstruction experiment, will be integrated into one paragraph.
> > >
> > > * We will add the "Stable Explanation" paragraph through \paragraph{} under the "Interpretable Explanation Space" in section 4.1. This paragraph is related to the experiment we answered earlier, and the manuscript to be added is as follows, possibly with some minor changes in expression.
> > >
> > > > Interpretable explanation models have to provide consistent explanations for similar inputs for stability [1]. This is because explanation loses reliability if different explanations are provided for similar inputs. To demonstrate the stability of our method, we measure the changes in distances, which was provided as an explanation for each prototype when inputs are distorted. For fairness of measurements, the distance change is normalized by distance before the input is distorted. As a result, our model shows higher stability for input distortions as illustrated in the figure. Specifically, there is a large difference in terms of average and median in the case of the baseline model, which means that there are explanations that provide significantly different distances when the inputs are distorted. In contrast, there is no large difference in terms of average and median in the case of our model, which means that explanation changes in proportion to input distortion. Based on our experiments, it is believed that our method provides a more reliable explanation to users than the baseline model.
> > >
> > > * We will add detailed setup related to the above experiment to the Appendix.
> > >
> > > ---
> > > **Reference:**
> > >
> > > [1] Alvarez-Melis, David and T. Jaakkola. “Towards Robust Interpretability with Self-Explaining Neural Networks.” NeurIPS (2018).

---

> ### Author Response · Authors · 2021-11-16
> **Response to Reviewer QGcu (2/3)**
>
>
> \
> **2. The current motivation for using the uncertainty layer seems to be quite weak.**
>
> Since VAE trains the model based on sampling, we hypothesized that the uncertainty in the classification process through the location in the latent space may increase by repeating the sampling process. Therefore, we tried to make the latent space as suitable for classification with VAE properties as possible. As a result, MNIST and FashionMNIST showed a 1% performance improvement. The motivation for using this uncertainty layer is briefly mentioned in Appendix A.5.
>
> While applying the uncertainty layer, the configuration of the latent space is well clustered by class so that it is more suitable for classification, shown in Figure 10 of the Appendix. If you look at the part where the uncertainty layer is not used in Figure 10 (b), you can see that the pink cluster and the surrounding cluster are a little bit mixed. On the other hand, if you look at the part where the uncertainty layer is used in Figure 10 (b), you can see that each cluster is neatly disentangled. Since the locations of the prototypes are determined for accurate classification in this latent space, the prototypes are positioned at a certain point that is better formed when the uncertainty layer is used as shown in Figure 11. This part was inspired by [1] to create certain data by shifting uncertain data from the latent space of VAE in a direction to reduce uncertainty.
>
> We aim to reduce the uncertainty of the classification process to make the configuration of the latent space and the location of the prototype more suitable for classification. Therefore, we think that the uncertainty layer can show advantages in other complex datasets. It seems that this was not well expressed in the paper, so we will mention the motivation in section 3.3 Using uncertainty.
>
> \
> **3. Some descriptions are either not clearly defined or proved.**
>
> In the case of inherently interpretable you mentioned, it means that the model directly generates a description of itself, as in the second paragraph of the Introduction. You can think of it as a concept compared to the post hoc interpretation method, which uses the second model to explain the first black-box model, which has recently exploded in the explainable AI field. Expressions such as inherently interpretable are being used in explainable AI-related papers such as [2], [3], [4], and [5].
>
> "Faithful" can be understood literally and can be replaced with words such as reliable. [2], which we cited, emphasizes the importance of inherently interpretable models because these post-hoc explanation methods are often not reliable, and can be misleading. [2] deals with the inherently interpretable model providing faithful explanations, which differs from post hoc explanation models. In the last line of the second paragraph of the Introduction, we tried to show this part, but our sentence seems to be lacking. We'll organize the citations in that part to make it easier to understand.
>
> In Abstract, we wanted to express that our model provides more faithful explanations than the previous post hoc explanation models because it developed as an inherently interpretable model rather than a comparison with the baseline. Because the word "faithful" is not said to be a measurable definition as you mentioned, we will omit the word **"more"** written in the abstract.

---

> ### Author Response · Authors · 2021-11-16
> **Response to Reviewer QGcu (1/3)**
>
> \
> We appreciate you very much for your detailed comments on our paper.
>
> \
> **1. What is the concrete problem this paper tries to solve?**
>
> Explanations must be both human-interpretable and reliable. Therefore, in the case of a prototype-based explainable model, since the distance and the prototype provided as an explanation must also be interpretable and reliable, here we questioned whether the explanation given as a distance explains the model well to users. For prototype-based explanation models, distances and prototypes are given as explanations, and when the user receives these explanations, they don't know why the model computes these distances. We think this is an important issue because users understand the model's prediction through the distance in a situation where distance and prototype are given as explanations. Therefore, by showing which features change through relationship, we tried to increase the understanding of model judgment by providing the user with what features the model influenced and outputted the distance. To check whether understanding of the model prediction increased, we checked whether users were able to respond to factors that influenced model prediction with and without additional relationships. Mechanical Turk experiments were conducted for the case of correct/wrong classification and the results are in 5.2.
>
> As mentioned earlier, in the prototype-based explanation method, it is very important that the distance value has a practical meaning because people understand the model's prediction through distance. We have determined that distance is not sufficient as an explanation for understanding the model unless it guarantees human perceived similarity. Therefore, we wanted distance to have an "intrinsic meaning" that can be visually expressed similarly to human perception.
>
> For example, if a Siamese cat is input, it is possible to closely predict the distance to a Sphynx cat prototype of a different shape (since distance does not guarantee similarity of shape). Our method in these situations allows users to see how distances are constructed through relationships when the users ask questions about the explanation provided by the model. This advantage works stronger in the case of misclassification. If the cat is mistakenly determined to be a dog, the model will provide an explanation of the distance to the dog prototype, but the user will not know why the model judged the distance to be close. However, our method shows the relationship between the cat and the dog prototype, so users can see what factors influence the model to compute distance. Therefore, our proposed model has an advantage in that it confirms the reliability of the explanation of distance.
>
> As you mentioned, it seems that the description of the part that expresses the strengths of our paper was insufficient in the introduction. We will put in the fourth paragraph of the introduction that users can check the composition of the distance through relationship and confirm the reliability of the explanation with distance. In addition, we will also include the fact that the problem of similar latent representations do not share similarities in appearance that prototype-based explanation methods were suffer was solved in terms of "intrinsic meaning" in the fifth paragraph of the introduction. Also, to help understand the purpose of the paper, we will summarize the part described with the simple example above and include it in the introduction. We appreciate your help in clarifying our strengths that we could not accurately describe in the paper, and we will upload a revised version within the discussion period.

---

### Author Response · Authors · 2021-11-21
**General Response**

We appreciate the reviewers for their constructive feedback. We updated the manuscript and uploaded a **revised version** of the paper to address the concerns of reviewers. The main improvements of our paper are as follows:

---

* In the introduction section, we revised the whole section to state the purpose of our paper and the problem that we are trying to solve. We also express the strengths of our paper for a clear understanding of our contributions. (Reviewer QGcu, HXZp, RHQ4)

* In the related work section, we have added relevant methods. (Reviewer QGcu)

* In section 3.3, we added the motivation of the uncertainty layer. (Reviewer QGcu)

* In the last paragraph of section 4.3, we added the limitation that visualization requires many training samples. (Reviewer RHQ4)

* In section 5.2, the description for the Mechanical Turk experiment has been revised to be clearer. (Reviewer QGcu, RHQ4)

* For clarifications, grammatical issues have been resolved and sloppy phrasings are revised to indicate more accurately. (Reviewer QGcu, RHQ4)

---

We hope our modifications and clarifications address most of the concerns about our paper. If any of our responses to individual reviewers or modifications are insufficient, please feel free to ask any additional questions.

---

### Decision · Program_Chairs · 2022-01-20

**Decision:**

Reject

**Comment:**

This paper proposes to create an explanation space to describe the relationships between input data and prototypes (and also between the prototypes themselves). It constructs such a space suing VAEs and conducts experiments to validate the effectiveness and interpretability of the method.

Strengths:
- The proposed method is interesting and intuitive

Weakness:
- Novelty of the idea is limited
- Missing experiment comparison with some important previous work
- Some claims are not well supported by the empirical results